# 3D chromatin maps of a brown alga reveal U/V sex chromosome spatial organization

Pengfei Liu [1], Jeromine Vigneau[1], Rory J. Craig[1], Josué Barrera-Redondo [1], Elena Avdievich[1], Claudia Martinho[1,3], Michael Borg [1], Fabian B. Haas [1], Chang Liu [2] & Susana M. Coelho [1] ✉

Nuclear three dimensional (3D) folding of chromatin structure has been linked to gene expression regulation and correct developmental programs, but little is known about the 3D architecture of sex chromosomes within the nucleus, and how that impacts their role in sex determination. Here, we determine the sex-specific 3D organization of the model brown alga *Ectocarpus* chromosomes at 2 kb resolution, by mapping long-range chromosomal interactions using Hi-C coupled with Oxford Nanopore long reads. We report that *Ectocarpus* interphase chromatin exhibits a non-Rabl conformation, with strong contacts among telomeres and among centromeres, which feature centromere-specific LTR retrotransposons. The *Ectocarpus* chromosomes do not contain large local interactive domains that resemble TADs described in animals, but their 3D genome organization is largely shaped by post-translational modifications of histone proteins. We show that the sex determining region (SDR) within the U and V chromosomes are insulated and span the centromeres and we link sex-specific chromatin dynamics and gene expression levels to the 3D chromatin structure of the U and V chromosomes. Finally, we uncover the unique conformation of a large genomic region on chromosome 6 harboring an endogenous viral element, providing insights regarding the impact of a latent giant dsDNA virus on the host genome's 3D chromosomal folding.

Sex chromosomes are unique genomic regions that evolved independently many times in different groups of eukaryotes. Three types of sex chromosome systems exist in nature, the well-described XX/XY and ZW/ZZ systems and the still elusive U/V systems, in organisms that express sex in the haploid stage of the life cycle[1–6]. Heteromorphic sex chromosomes (Y, W, U, V) have evolved repeatedly in diverse eukaryotic species. Suppression of recombination between X and Y (or Z and W, or U and V) chromosomes usually leads to a range of genomic modifications in these regions, including degeneration of the non-recombining chromosome, accumulation of repeats, and gene decay via

accumulation of deleterious mutations[7,8]. Repeats pose the largest challenge for reference genome assembly, and centromeres, subtelomeres, and repeat-rich sex chromosomes are typically ignored in sequencing projects. Consequently, complete sequence assemblies of heteromorphic Y, W, U, and V sex chromosomes have only been generated across a handful of taxa[9–16], and most of the information is fragmentary even at the linear sequence level. Moreover, despite the key association between the three-dimensional (3D) structure of chromatin and gene regulation[17,18], we lack critical information regarding chromatin landscapes and nuclear 3D organization of sex chromosomes

[1]Department of Algal Development and Evolution, Max Planck Institute for Biology Tübingen, Tübingen, Germany. [2]Institute of Biology, University of Hohenheim, Stuttgart, Germany. [3]Present address: School of Life Sciences, Division of Plant Sciences, University of Dundee, At James Hutton Institute, Errol Road, Invergowrie, Dundee, UK. ✉e-mail: susana.coelho@tuebingen.mpg.de

within the nuclear space, and how chromatin folding is associated with the sex-specific gene expression underlying sexual differentiation.

Genome folding generally involves hierarchical structures ranging from chromatin loops to chromosome territories[19]. The best-known 3D chromatin organization units are topologically associating domains (TADs), which show a self-interacting pattern with strongly interacting boundaries in Hi-C contact maps of animal genomes[20]. Genome architectural proteins, such as CTCF (CCCTC-binding factor) and cohesin, bind strongly to DNA anchor sites and mediate the formation of chromatin contact domains through loop extrusion[21]. In addition to TADs, structural units called compartmental domains have been demonstrated in animals (e.g.,[22]). Compartmental domains are closely associated with local chromatin states and preferentially interact with other compartmental domains of similar chromatin states, contributing to the establishment of the 3D architecture for a given genome[19,22]. Plant genomes also frequently exhibit a higher-order 3D chromatin organization. TADs or TAD-like structures have been described in several plant species[23], although their genomes do not encode CTCF homologs[24]. In contrast to animal TADs that have sharp and well-delineated boundaries on Hi-C maps, plant TADs exhibit less pronounced boundaries due to weaker chromatin insulation[25]. In contrast, *Arabidopsis* (*Arabidopsis thaliana*) lacks plant TADs[26]. The absence of TADs in the *Arabidopsis* genome is likely related to its small size, high gene density, and short intergenic regions. However, chromatin loops and A/B compartments are present in *Arabidopsis* (e.g.,[27,28]), and small structural units within 3D chromatin architecture have been recently described[29].

Here, we generated 2 kb-resolution maps of the male and female haploid genomes of the brown algal model *Ectocarpus* and examined the 3D chromatin structure of autosomes compared to U and V sex chromosomes. The *Ectocarpus* life cycle involves an alternation between diploid and haploid generations, with sex being determined in the haploid stage of the life cycle by U (female) and V (male) sex chromosomes[5]. Therefore, this model organism provides the opportunity to investigate the U/V sex chromosome organization in comparison to autosomes. Our near-complete assembly of the *Ectocarpus* genome (*Ectocarpus* V5) offers an improved reference genome and allows us to define and characterize the centromeric and subtelomeric sequences in this organism. We found that interphase chromatin is organized in a non-Rabl configuration, with telomeres and centromeres of all 27 *Ectocarpus* chromosomes clustering together in the 3D nuclear space. We reveal that the 3D structure of *Ectocarpus* chromatin is highly streamlined, not organized into TADs, and A and B chromatin compartments are mainly defined by H3K79me2 deposition and depletion of activation marks. We then focus on the 3D structure of the *Ectocarpus* U and V sex chromosomes to show that in both sex chromosomes, the SDR spans the centromere, and is insulated from the rest of the chromosome. We found no overall differences in the 3D chromatin organization between male (V) and female (U) chromosomes but both have different 3D organization compared with autosomes. Finally, we uncover the distinctive conformation of a genomic region on chromosome 6 harboring a giant endogenous viral element (EVE), giving insights into the interplay of dsDNA viruses with the chromatin environment in the host.

## Results

### A near complete assembly of the male and female haploid genome of *Ectocarpus*

A complete assembly of the *Ectocarpus* genome has been challenging mainly due to the presence of highly repetitive regions, which short-read Illumina sequencing, low coverage Hi-C, and Sanger sequencing could not hitherto successfully resolve. The published version of the *Ectocarpus sp. 7* reference genome (V2) contains 28 pseudo-

chromosomes spanning 176.99 Mb, with 17.97 Mb of unplaced contigs, a contig-level N50 of 33 kb, and a total of 11,588 gaps[30]. Here, we combined Oxford Nanopore Technologies (ONT) long reads and Hi-C sequencing techniques to achieve near-complete assemblies of both the haploid male and female genome of *Ectocarpus sp. 7* (Fig. 1A and Supplementary Data 1, 2).

The ONT long reads were obtained separately from male and female siblings (Ec32m, Ec25f, Supplementary Figs. 1, 2), totaling 11 Gb and 20 Gb of data, respectively. ONT long-reads were complemented with Hi-C data, encompassing 822 million pairs of sequences (~135 Gb) at a sequencing depth of 635x for the male, and ~444 million pairs (~73 Gb) and 338x coverage for the female (see "Methods" for details) (Supplementary Fig. 1 and Supplementary Data 3). The *Ectocarpus* V5 male genome assembly has an N50 of 7.0 Mb and a total size of 186.6 Mb. Chromosome 28 from V2 is now part of chromosome 4, bringing the total number of chromosomes to 27, with sizes ranging from 4.52 Mb to 10.73 Mb. Similarly to what was done for the *Ectocarpus* V2[30], we added the female SDR (size 1.55 Mb) to this male genome in order to obtain a final *Ectocarpus* V5 reference genome (Supplementary Fig. 1).

The genome is highly contiguous: out of the 27 assembled chromosome models, most are gapless, and only six chromosomes have 10 gaps in total (Fig. 1A and Supplementary Data 4). The accuracy of Hi-C-based chromosome construction was evaluated manually by inspecting the chromatin contact matrix at 100 kb resolution, which exhibited a well-organized interaction contact pattern along the diagonals within each pseudo-chromosome (Fig. 1B).

Telomeric regions were almost entirely absent from earlier genome assemblies, although a putative telomere bearing the repeat (TTAGGG)n was observed[31]. In our *Ectocarpus* V5 assembly, 43 of the 54 telomeric regions are fully resolved, and twelve of the 27 pseudo-chromosomes correspond to a telomere-to-telomere assembly (Fig. 1A and Supplementary Fig. 3). On all but three of the resolved telomeric regions, we observed a specific satellite repeat adjacent to the telomeric repeats. The satellite features a repeated monomer of ~98 bp and is almost exclusively found at the sub-telomeres, where it forms arrays that range from only a few to more than 100 copies. Notably, the telomeric motif TTAGGG is present in three independent locations within each satellite monomer (Supplementary Fig. 4). Similar sub-telomeric organizations have been observed in several species, including the green alga *Chlamydomonas reinhardtii*, where the telomere-like motifs present within the sub-telomeric satellites are hypothesized to serve as seed sequences that facilitate telomere healing following DNA damage[32]. Eight of the V5 chromosome arms terminated in the subtelomeric repeat, leaving only four chromosome extremities for which the assembly failed to reach either the subtelomere or telomere (Fig. 1A).

Ribosomal DNA (rDNA) arrays were also poorly resolved in previous assemblies. The V5 assembly revealed a single major rDNA array located within an internal region of chromosome 4 (position: 269–274 Mb), which features six rDNA repeats before collapsing the assembly due to high levels of repeat content. The 5S rDNA gene is linked to the main rDNA unit (18S-5.8S-26S), as previously reported in many brown algae and Stramenopiles[33]. Considering an ONT read coverage fold change of approximately 20 between the rDNA array and the flanking regions, we estimate that the rDNA array may consist of >100 rDNA repeats, spanning ~1 Mb.

The total repeat content of the assembled chromosomes is estimated to be 29.8% (Supplementary Data 4). 13.75 Mb of additional sequence could not be assembled into chromosomes and remains unplaced in the V5 assembly. These sequences are highly repetitive (74.3% repeats) and presumably include heterochromatic regions that correspond to some of the assembly gaps or incomplete chromosome ends. Longer reads or alternative technologies will be required to achieve complete assembly of these complex regions.

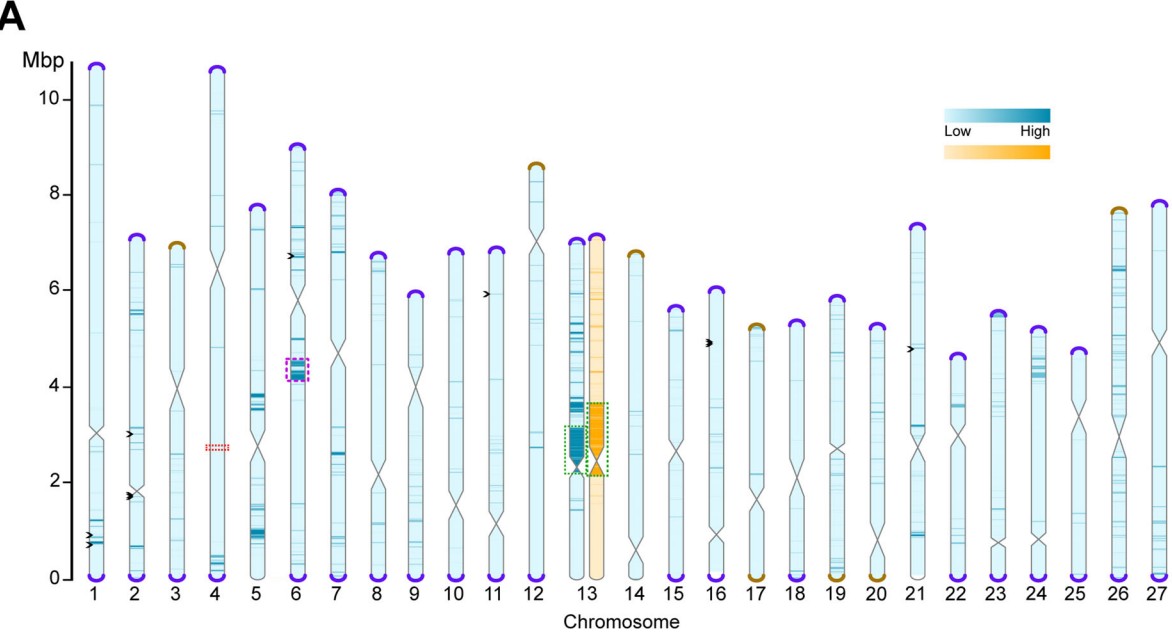

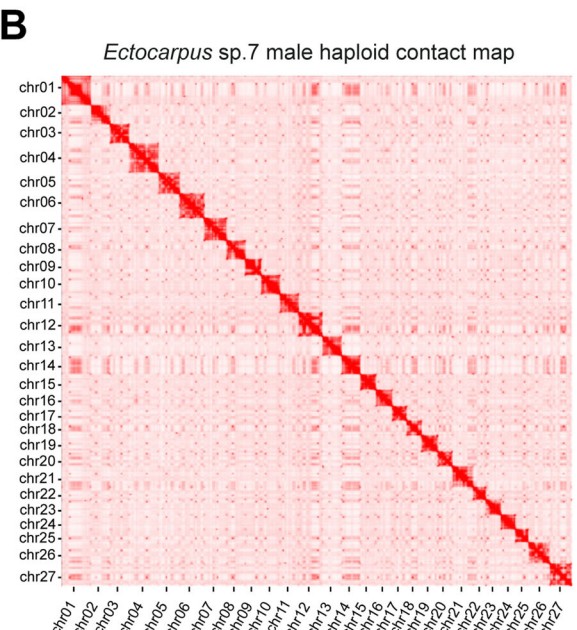
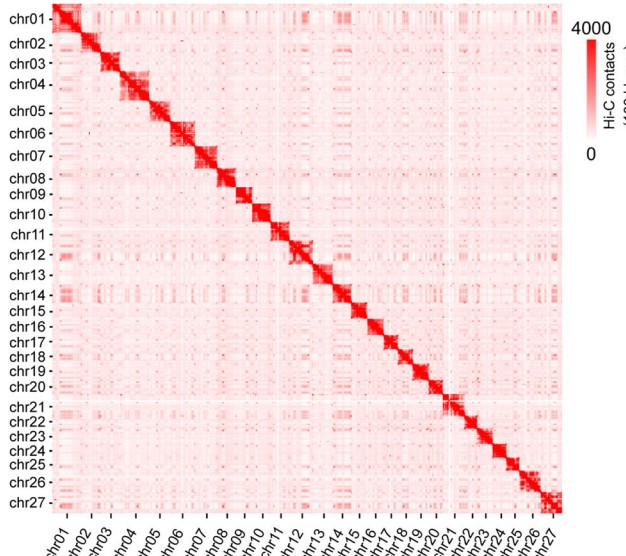

**Fig. 1 | *Ectocarpus sp. 7* whole-genome assembly. A** Schematic representation of the near telomere-to-telomere assembly of the 27 *Ectocarpus sp. 7* chromosomes, in haploid male (blue) and female (orange). Telomeres are represented as violet caps, sub-telomeres in brown. Centromeric regions are represented by the constrictions in the center of the chromosomes. The chromosomes are filled by variant density between the male and female haploid genomes used for the assembly (darker color means more differences). Violet dotted boxes represent the genomic region where a dsDNA virus is inserted, green dotted boxes represent the SDRs, and the red dotted box shows the rDNA array. Black arrowheads depict gaps. See methods for details. **B** Normalized genome-wide Hi-C contact map showing frequencies of pairwise 3D genome contacts at a 100 kb resolution in the male and female haploid genomes. The stripes seen in the contact map indicate regions of high contact frequency, corresponding to A/B compartments where regions within the same compartments interact more frequently with each other than with regions in other compartments. The dots scattered across the contact map represent specific loci that have higher contacts than the rest of the genome, often suggesting interactions between telomeres and centromeres or loops between these regions.

Since the V2 genome had a high-quality gene annotation, we performed a liftover of the V2 gene models to the *Ectocarpus* V5 genome. Out of the 18,412 V2 gene models, 18,278 could be lifted, and the remaining were mostly located on an unassigned scaffold in the V2 assembly. Genome completeness was quantified by BUSCO[34]. Two database sets were used, Eukaryota (255 core genes) and stramenopiles (100 core genes). Of the 255 core eukaryotic BUSCO genes, the V5 reference assembly contains 226 (88.7%) complete BUSCO genes. This represents a gain of 8 genes (+ 3.2%) compared to the V2 genome. The stramenopiles dataset resulted in 93% BUSCO completeness (increased by 1%; Supplementary Data 5).

### The *Ectocarpus* 3D chromatin architecture

To explore the 3D chromatin architecture of *Ectocarpus*, we mapped male and female Hi-C reads back to the V5 assembly (Supplementary Fig. 1 and Supplementary Data 3). Biological replicates were highly

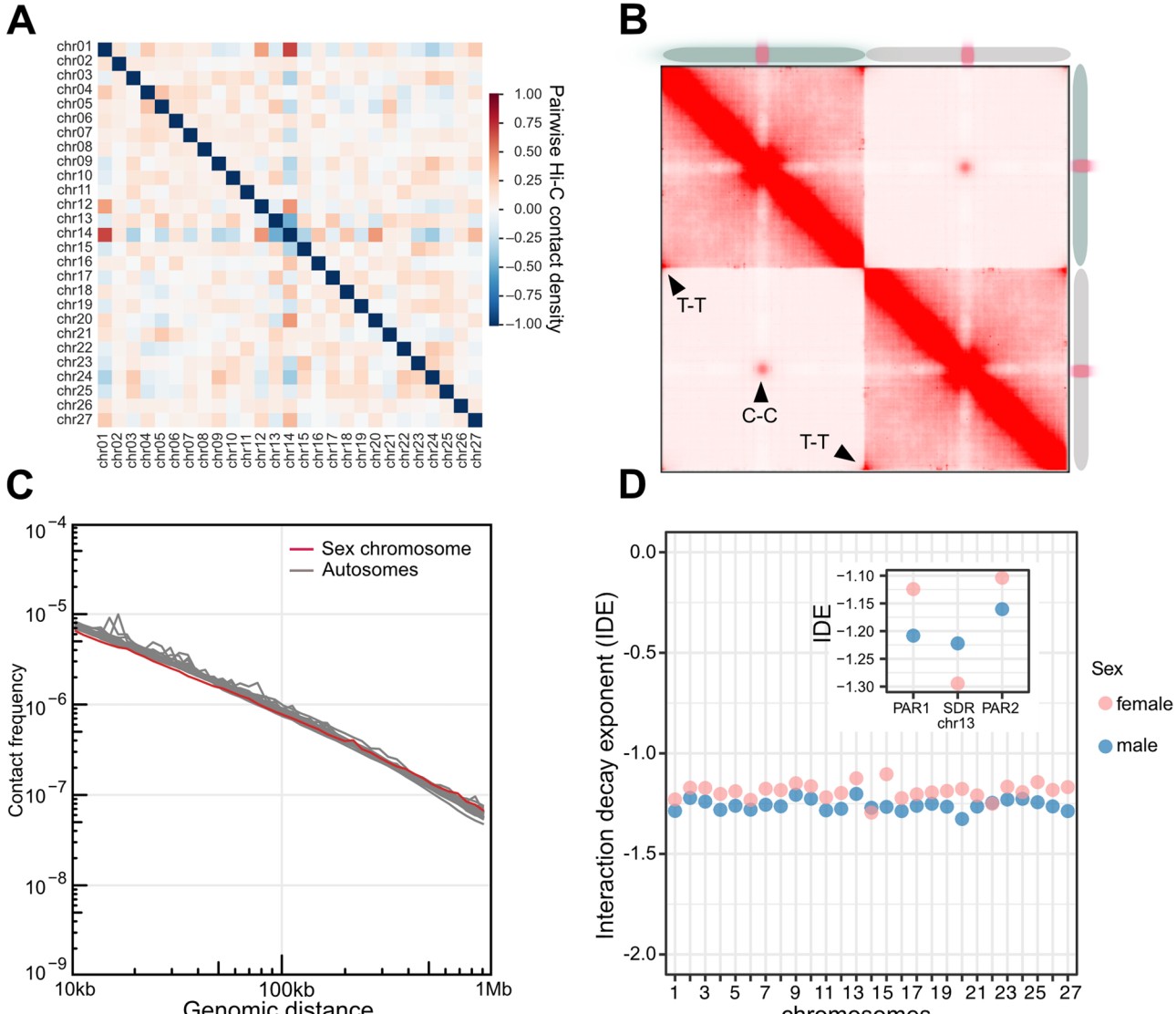

**Fig. 2 | 3D chromatin architecture of *Ectocarpus* revealed by Hi-C data. A** Pairwise averaged log-transformed observed/expected inter-chromosome contacts of *Ectocarpus* male at 10 k resolution. This scale represents the relative density of interactions between different chromosomes, with darker colors representing higher contact frequencies compared to what is expected by chance. **B** Analysis of aggregated intra- and inter-chromosomal contacts (Aggregate Chromosomal Analysis, ACA[104]), where individual chromosomes are linearly transformed to have the same length, and the centromere is placed at the center. The adjusted chromosomes are subsequently used to compute average intra- and inter-chromosomal contacts. The analysis shows how centromeres and telomeres of these chromosomes interact both within themselves and with each other. T-T: telomere to telomere interactions; C-C: centromere to centromere interactions. **C** Global folding patterns of each of the male *Ectocarpus* chromosomes reflected by contact frequency as a function of genomic distance (Ps). **D** IDEs of each autosome and sex chromosome region in *Ectocarpus* male and female. Normalized Hi-C matrices at a resolution of 10 kb at a distance range of 10 kb to 500 kb were used to calculate IDEs.

correlated (Pearson $r = 0.96$ and $r = 0.94$ for male and female samples, respectively, Supplementary Fig. 5), therefore, replicates were combined for downstream analysis to produce sex-specific high-resolution maps. We obtained 188.8 and 134.8 million interaction read pairs for male and female *Ectocarpus*, respectively, reaching a 2 kb resolution for each of the sexes.

In animals and plants, chromosomes are hierarchically packed in the nuclear space, and each occupies discrete regions referred to as a chromosome territory (CT)[23,35]. Chromosomal territories were detected in *Ectocarpus*, reflected by strong intra-chromosomal interactions and clear boundaries between chromosomes (see Fig. 1B). The global Hi-C maps of male and female *Ectocarpus* show no noticeable differences among autosomes, suggesting that the overall chromosomal territory organization is highly similar between the sexes. In addition, both the male and female sex chromosomes do not display any distinct intra- or inter-chromosomal contact patterns that differentiate them from autosomes. Therefore, the *Ectocarpus* genome folding on a broad chromosomal level appears to be consistent across both sexes and all chromosomes. We found a significant enrichment of inter-chromosomal interactions involving chromosomes 1, 12, 14, 20, and 27 (Fig. 2A), suggesting a propensity for these chromosomes to establish stronger contacts compared to others. Furthermore, strong contacts among telomeric regions of different chromosomes, as well as contacts among centromeric regions (see below), were widespread on the Hi-C map (Fig. 2B).

Next, we computed each chromosome's chromatin contact probability as a function of genomic distance to examine *Ectocarpus* chromosome packing patterns. As expected, we observed a decline in contact frequencies as genomic distances increased (Fig. 2C). Next, Interaction Decay Exponents (IDEs), which describe how fast

interaction frequencies drop with increasing physical genomic distance, were computed to characterize chromatin packaging[36,37]. We found that for each of the *Ectocarpus* chromosomes, interaction frequencies decayed in similar power-law functions with IDE values between 10 kb and 500 Kb (Fig. 2D). However, the IDE values in SDRs and PARs of sex chromosomes showed noticeable variation, suggesting differences in local chromatin packing in these regions (see below).

One prominent feature of animal and plant genomes is the organization of chromatin into TADs, characterized by preferential contacts between loci inside the same TAD and strong insulation from loci in adjacent TADs[20,38]. In some cases, TADs can promote enhancer-promoter contacts important for gene expression[39]. Intriguingly, we did not observe conspicuous TADs patterns in any of the *Ectocarpus* chromosomes upon zooming into the Hi-C map (Supplementary Fig. 6). Note that *Ectocarpus* has a similar genome size to the land plant *Arabidopsis*, which also does not exhibit classical TAD structure, but rather TAD-like domains that are moderately insulated from flanking chromatin regions[26,40] and are considered as an outlier species concerning plant TAD formation[41].

## A/B compartment dynamics in males versus females

Spatially distinct nuclear compartments are a prominent feature of 3D chromatin organization in eukaryotes[36]. A/B compartments, which generally correlate to active and repressed chromatin, respectively, can be identified with the first eigenvector (EV1) generated from the principal components analysis of the correlation heatmap (PCA)[36]. We applied PCA to individual chromosome's Hi-C maps normalized at 10 kb bin size to identify the two spatial compartments (Fig. 3A). The compartment that displayed stronger inter-chromosomal chromatin contacts was called 'A', whereas the 'B' compartment had lower inter-chromosomal contacts (Fig. 3B). Interestingly, the genomic regions bearing the centromeres corresponded to the A compartment (Fig. 3A, B). Further PCA analysis on the A compartment indicated that centromeres formed distinct sub-compartments, which were spatially separated from the rest of the A compartment regions (Supplementary Fig. 7).

Although different chromosomes had different proportions of compartment A and B, we noticed that the U and V sex chromosomes (chromosome 13) exhibit large stretches of regions associated to B compartment (Fig. 3B), suggesting they have a distinct overall configuration compared to autosomes (see below).

The *Ectocarpus* genome has been reported to have various histone post-transcriptional modifications (PTMs) associated with gene transcriptional activities[42,43]. We therefore asked whether chromatin associated with different A/B compartments exhibited different histone modification profiles. To this end, we used published epigenomic datasets for a range of histone PTMs from the same strains (Ec560, Ec561)[44] and mapped the ChIP-seq datasets to our V5 genome. We then examined the enrichment of each histone PTM specifically associated with genes in each A or B compartment (see "Methods"). We found that for both male and female genomes, the histone PTMs associated with active gene expression, such as H3K4me3, H3K9ac, H3K27ac and H3K36me3, were significantly enriched in the A compartment (Fig. 3C) although more modestly so for H3K36me3. Conversely, peaks of H3K79me2, a histone mark associated with repressed chromatin in *Ectocarpus*[42,43], were only marginally enriched in the B compartment albeit significantly (Fig. 3C). Furthermore, genes located within A compartment regions exhibited higher expression levels than those in the B compartments (Fig. 3D). Note that canonical repressive (heterochromatin) marks such as H3K9me2 and H3K27me3 are absent in *Ectocarpus*[42,43], with higher transcript abundance correlating more strongly with the gradual acquisition of activation-associated marks (H3K9ac, H3K27ac, H3K4me3 and H3K36me3[43]. Our observations thus suggest that transcriptional states appear to be the main driver of chromatin organization patterns in *Ectocarpus*, unlike what has been

reported in other multicellular eukaryotes where the A/B compartments are enriched with euchromatic and heterochromatic chromatin, respectively[23,36]. Alternatively, *Ectocarpus* might harbor other yet-to-be-characterized repressive marks that could influence the organization of A/B compartments.

The A/B compartment assignment of the male and female *Ectocarpus* genomes was highly similar; nonetheless, 5.3% of the *Ectocarpus* chromatin exhibited different A/B compartment identities in male and female Hi-C maps (Fig. 3E). In animals, compartment status and boundaries may change during cell differentiation and correlate with changes in gene expression profiles[45]. We therefore investigated whether such changes in compartment annotation were associated with the expression patterns of sex-biased genes (SBG), i.e., genes that show a significant change in expression in males versus females[46,47]. We used RNA-seq datasets[44] and identified 2069 SBGs (see "Methods" for details, Supplementary Data 7). Depending on the expression preference, these SBGs were further annotated as male- and female-biased genes (MBGs and FBGs), respectively. SBGs were, however, not enriched in the regions where A/B compartment identity changed (Supplementary Data 8, Chi-square test $p = 0.0649$). MBGs in males were upregulated when in compartment A compared to FBGs, and the opposite was true in females (Fig. 3F), and we noticed that whilst MBGs in males still show greater expression associated to the A compartment, the overall expression levels were higher, regardless of the compartment. Therefore, the patterns of expression of SBGs were correlated with their association with histone PTMs and to the specific 3D chromatin organization in males versus females.

## U and V sex chromosomes and autosomes adopt distinct conformations

The sex-specific high-resolution genomic maps were then used to compare the sub-nuclear 3D genomic architecture of the U and V sex chromosomes. The U and V sex-specific regions (SDR) have been identified and characterized previously[9,48,49], but their largely repeat-rich nature has prevented their full assembly. In the *Ectocarpus* V5, the V and U chromosomes had a total length of 7.16 Mb and 7.23 Mb respectively (see Fig. 1). In *Ectocarpus*, U and V are largely homomorphic with a small region that is non-recombining (SDR) and therefore largely divergent between male and female[9,49] (Supplementary Fig. 8). The male and female SDRs of the *Ectocarpus* V5 genome feature no gaps. We also noticed that compared to the V2, the female SDR has increased in physical size. This was mainly due to the addition of repeats in the new assembly (V2 had 34.7% of repeats and V5 68.3% of repeats in the U-SDR). The small SDRs are flanked by large pseudoautosomal regions (PARs), which recombine at meiosis[9,50]. Structural analysis using our new assembly confirmed that the U and V sex chromosomes display unique characteristics compared with autosomes, including lower GC content, higher repeat content, lower gene density[9,50], and a largely repressive chromatin landscape[43] (Fig. 4A and Supplementary Data 4). We then used the 2 kb resolution Hi-C map to investigate the 3D structure of the sex chromosomes in the *Ectocarpus* nucleus. Intriguingly, the U and V sex chromosomes exhibited a distinct 3D architecture compared to autosomes, with their central SDRs both being insulated from the flanking PAR regions (Fig. 4B, see also Fig. 2D), with high intra-chromosomal contacts in the 3D space. We also noticed that both U and V SDRs spanned the centromeres (Fig. 4B).

## *Ectocarpus* centromeres are distinguished by specific LTR retrotransposons

To determine the structure and precise locations of the *Ectocarpus* centromeres, we analyzed the sequence characteristics of the chromosomal regions delineated by centromere-to-centromere interactions (see Figs. 2B, 3A, B). Regional centromeres vary extensively among eukaryotes, with common structures including short non-

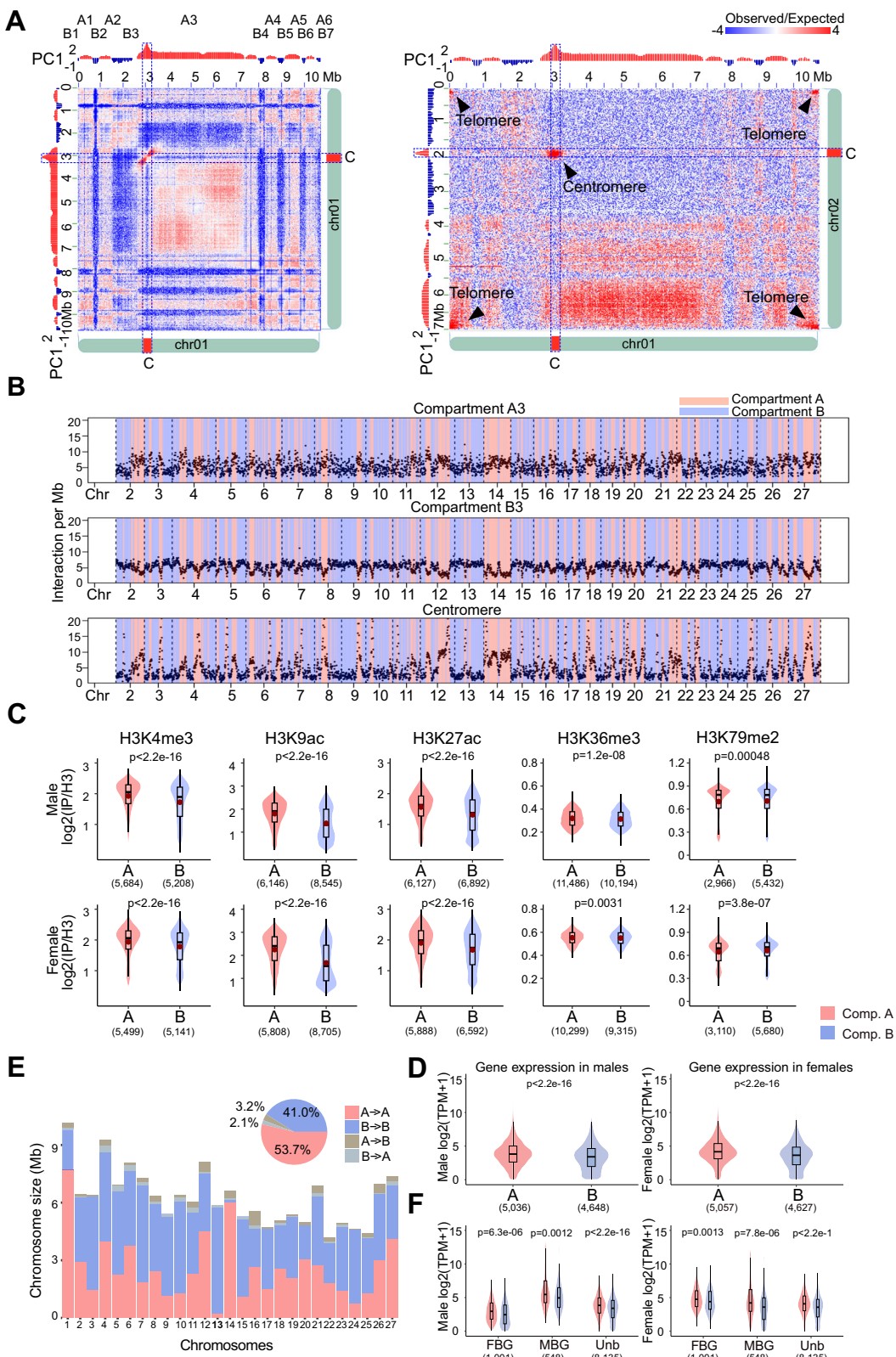

repetitive AT-rich regions, transposon-rich regions spanning tens to hundreds of kilobases, and megabase-scale satellite arrays[51]. We first searched for any specific repeat families that were (i) enriched in the putative centromeric regions, and (ii) common to all chromosomes. This revealed two retrotransposon families that are almost exclusively restricted to a single highly localized, gene-poor, and repeat-rich region on each chromosome (Fig. 5A and Supplementary Fig. 9). The

most abundant of the two elements is a 6.6 kb *Metaviridae* (i.e., *Ty3/Gypsy*) long terminal repeat (LTR) retrotransposon, which encodes Gag and Pol on a single open reading frame of 1699 aa and can be found as full-length copies flanked by 4 bp target site duplications (Fig. 5B). The second is presumably a related LTR element, although it is only present in degraded fragments, and we were unable to recover an internal protein-coding region. The two retrotransposon families

**Fig. 3 | High-resolution contact probability map reveals the higher-order organization of the *Ectocarpus* genome. A** Compartment A/B annotation based on principal component analysis. PC1 stands for the first principal component. The right panel shows inter-chromosomal contact patterns of A/B compartment regions between chromosomes 1 and 2. **B** Inter-chromosomal contacts of selected chromosome 1 regions with other chromosomes. The plots describe inter-chromosomal contacts belonging to the compartments A3 (top), B3 (middle), and centromere regions (bottom). The A/B compartment annotation of individual chromosomes is indicated with different colors. **C** Comparison of histone modifications, represented as log2(IP/H3) of regions enriched with selected histone marks. For each histone mark, the enriched regions are grouped according to the A/B compartments of *Ectocarpus* male and female. The mean value of log2(IP/H3) is represented by a red dot in each boxplot. Numbers in brackets represent the number of peaks of the corresponding histone ChIP-seq data. **D** Levels of gene

expression in compartments (**A**) and (**B**) in males and females, represented as log$_2$(Transcripts Per Million + 1). *p*-values represent Wilcoxon tests. Numbers in brackets represent the number of genes. **E** Lengths of conserved and switching A/B compartment regions in male and female Ectocarpus genomes. "X - > Y" indicates compartment annotation in males ("X") and females ("Y"). The pie chart indicates pooled data from all chromosomes. **F** Expression of sex-biased genes (SBG) in compartment A/B regions. MBG: male-biased gene; FBG: female-biased gene; Unb, unbiased gene. Numbers in brackets represent the number of genes. The lower and upper hinges of the box correspond to the first and third quartiles (the 25th and 75th percentiles). The upper whisker extends from the hinge to the largest and smallest values no further than 1.5x IQR from the hinge (Inter-Quartile Range, distance between the first and third quartiles). *P*-values represent a two-sample Wilcoxon rank sum test.

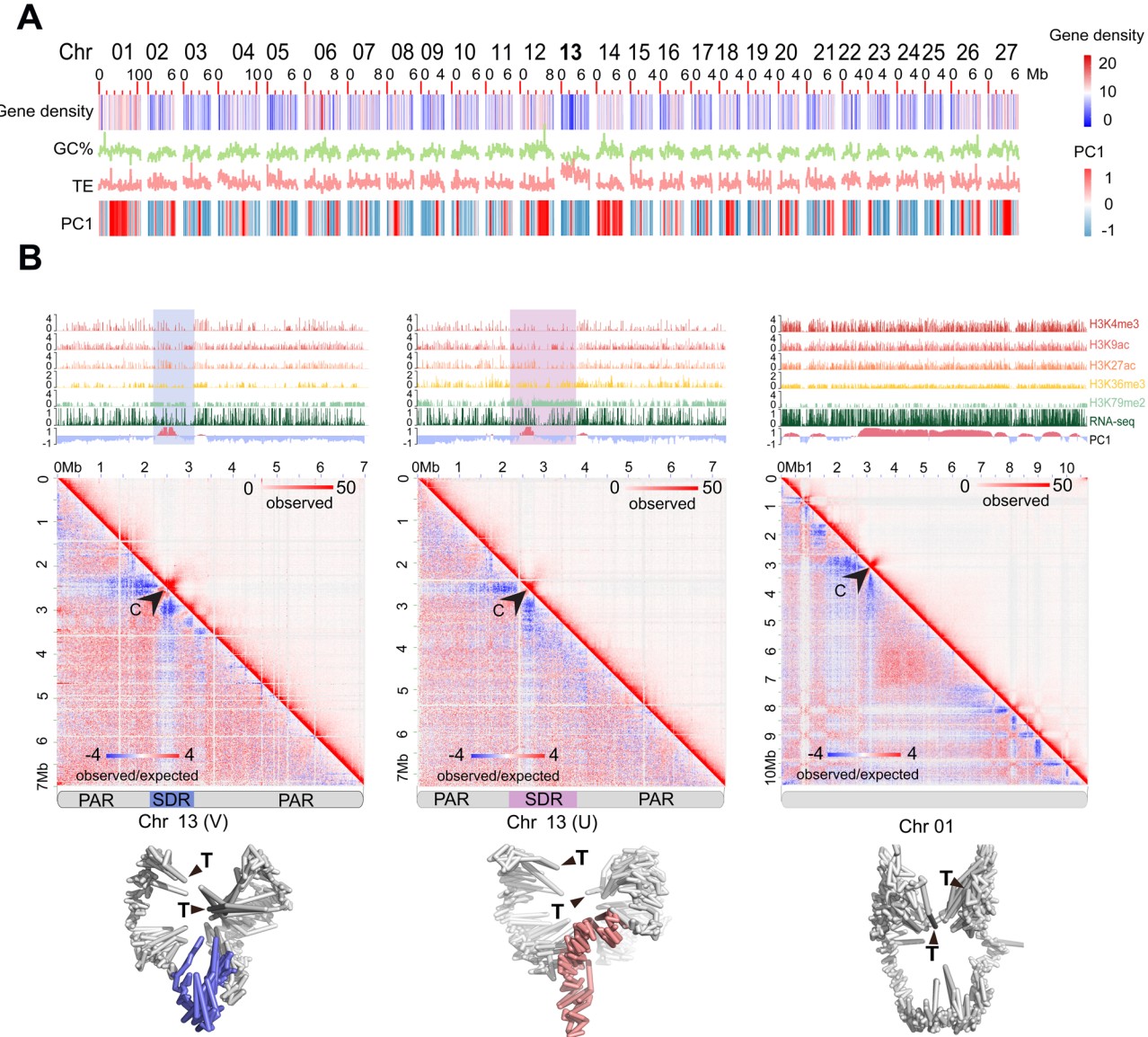

**Fig. 4 | U and V sex chromosome 3D architecture. A** Plot showing gene density, GC content, and TE density in 100 kb windows and compartment A/B (PC1, red indicates positive values corresponding to compartment A, and blue indicates negative values corresponding to compartment B) in 10 kb windows across the *Ectocarpus* chromosomes. Chromosome 13 is the sex chromosome. **B** Hi-C map and simulated 3D configurations of sex chromosomes at 10 k resolution, employing a

maximum likelihood approach, chromosomal structures were constructed from Hi-C data with a default setting of 3D Max[105]. SDRs in the simulated male and female chromosomes are colored in blue and red, respectively, and telomeres are labeled with black triangles. In each panel, the black arrowhead indicates the centromere. The tracks above each Hi-C map show A/B compartment annotation (PC1), gene expression (RNA-seq), and various histone modification ChIP-seq.

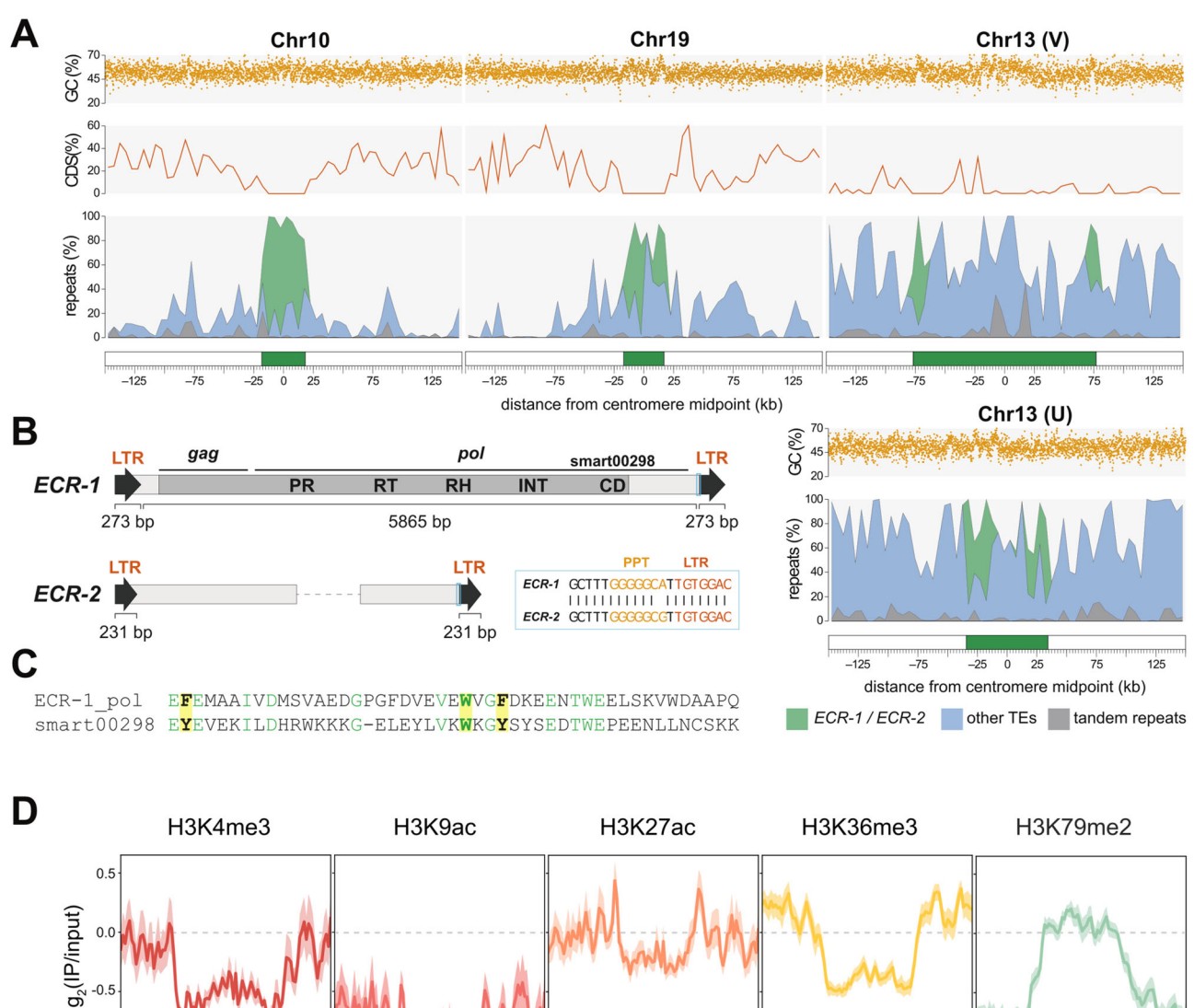

**Fig. 5 | *Ectocarpus* centromeres and centromere-specific retrotransposons.** **A** The centromeric regions of select chromosomes and the *ECR* retrotransposons. Putative centromeres and flanking regions for four chromosomes, including the U chromosome from the female genome assembly. The centromere (green box) is defined as the region from the first to the last copy of *ECR* elements. The repeats panel is shown as a stacked area plot, and the percentage of each repeat type is plotted in 5 kb windows. Coding sequence (CDS) density is plotted in 5 kb windows, and GC content is plotted in 100 bp windows. For all chromosomes, see Supplementary Fig. 9, and for genomic coordinates, see Supplementary Data 6. **B** Schematics of the *ECR* retrotransposons. The light blue boxes highlight the conserved region between *ECR-1* and *ECR-2*, and a partial alignment of this region is shown (PPT = polypurine tract). Only 5′ and 3′ fragments of *ECR-2* were recovered, and the dashed line represents a protein-coding sequence that is presumably missing. The domains shown on the *ECR-1* protein are: PR = protease, RT = reverse transcriptase, RH = RNaseH, INT = integrase, CD = chromodomain. **C** Alignment of *ECR-1* chromodomain and SMART chromodomain curated model (smart00298). Conserved amino acids are colored green, and the three aromatic amino acids that are responsible for the recognition of histone-methylated lysines are highlighted in yellow. **D** Histone mark signal (log2(IP/input)) in the putative centromeres and the surrounding regions (30 kb). Profiles of histone marks around the centromeres. The solid line represents the $\log_2$ of the ChIP-seq signal relative to the input, while the shading represents the 95% confidence interval (CI) of the center for the error bands. Heatmaps showing the chromatin state of each centromere using both uniquely and multi-mapped reads are shown in Supplementary Fig. 10.

share a ~ 64 bp region of homology that includes the polypurine tract that immediately precedes the 3′ terminal repeat (Fig. 5B). We name these elements *ECR-1* and *ECR-2* for *Ectocarpus* Centromeric Retrotransposon.

Notably, the *ECR-1* polyprotein features a C-terminal chromodomain fused to the integrase domain (Fig. 5B, C). Chromodomains recognize and bind histone-methylated lysines via a cage tertiary structure that is formed by three aromatic residues[52], all of which are conserved in the *ECR-1* polyprotein (Fig. 5C).

Defining the putative centromeres as the region between the first and last *ECR* element, lengths range from only 6.8 kb on chromosome 25 (essentially a single copy of *ECR-1*) to 153.9 kb on the male chromosome 13 (i.e., chromosome V), with a median of 38.6 kb (Supplementary Data 6). On average per centromere, 33% of bases are contributed by *ECR-1*, 7.3% by *ECR-2*, and 34% by other interspersed repeats that are not exclusive to these regions. Tandem repeats constitute only 3.5% of the putative centromeres, relative to 6.7% elsewhere in the genome. Although genes were generally absent from

these regions, certain chromosomes feature a small number of genes distributed among the *ECR* copies, including chromosome V, (Fig. 5A). Furthermore, only one of the four SDR genes that are contained within the V centromere was sex-linked in the last brown algal common ancestor (Ec-13_001830), while the others became sex-linked later during the expansion of the SDR in the Ectocarpales (Ec-13_001830.1) and in the expansion that is exclusive of *Ectocarpus* (Ec-13_001870.1 and Ec-13_001890.1)[49]. The GC content of the putative centromeres (52.7%) is only marginally lower than the rest of the genome (53.5%). However, this is partly driven by the GC content of *ECR-1* (58.2%), and several chromosomes do feature short AT-rich sequences within the putative centromeres (e.g., chromosome 19, Fig. 5A). As expected, following their evolutionary independence, the putative centromere of the female U chromosome differs substantially in length and composition relative to the V chromosome.

To further characterize the *Ectocarpus* putative centromeres, we analyzed the associated chromatin pattern using ChIP-seq data (Fig. 5D and Supplementary Fig. 10). Despite their assignment to compartment A, the putative centromeres exhibit a slight enrichment with the H3K79me2 mark in comparison to the surrounding genomic regions in the male SDR. In contrast, the histone marks associated with active genes (H3K4me3, H3K9ac, H3K27ac, and H3K36me3) were strongly depleted within the putative centromeres but strongly enriched in the surrounding regions, consistent with their compartment assignment and the presence of flanking genes. Interestingly, on a few chromosomes, the H3K79me2 pattern extends beyond the boundaries of the *ECR* elements. This observation holds true when using a different mapping method (removing multi-mapping reads) for chromosomes 16, 22, and V (Supplementary Fig. 10A, B), all of which have flanking regions that are highly enriched with interspersed repeats (TEs).

## An inserted (endogenous) viral element exhibits a unique chromatin conformation

Marine filamentous brown algae of the order Ectocarpales frequently carry endogenous giant viruses with large double-stranded DNA genomes[53]. *Ectocarpus sp. 7*, in particular, has been shown to harbor such type of endogenous viral element inserted in chromosome 6, derived from the *Ectocarpus* phaeovirus EsV-1[31,54,55]. We confirmed the presence of an endogenous viral element (that we name Ec32EVE) localized within chromosome 6 in our V5 *Ectocarpus* genome (Fig. 6A, see also Fig. 1A,). The Ec32EVE is 399 kbp long, contains 199 genes, and is covered with a large domain of the repression-associated mark H3K79me2 previously shown to be associated with the silencing of transposable elements in *Ectocarpus*[42,43]. The Ec32EVE region exhibits a depletion of activation-associated histone marks H3K4me3, H3K9ac, H3K27ac, and H3K36me3 (Fig. 6A). Consistent with this heterochromatic landscape, RNAseq analysis showed negligible expression throughout the entire Ec32EVE region (Fig. 6A), highlighting the silent nature of the potentially coding regions within the endogenous viral element. The chromosome 6 Hi-C map further revealed high levels of compaction and insulation, associated with the viral insertion region (Fig. 6A). Remarkably, the Ec32EVE region displayed strong long-range contact with telomeres in the nuclear 3D space (Fig. 6B). We asked whether this observation was related to the highly heterochromatic nature of the Ec32EVE region. However, other genomic regions equally marked with long stretches of H3K79me2 did not necessarily cluster in 3D with telomeres (Supplementary Fig. 11). It appears, therefore, that the Ec32EVE insertion, rather than the chromatin state of this region per se, is implicated in the unique 3D structure of this region.

## Discussion

High-quality and complete reference genome assemblies are fundamental for the application of genomics to a range of disciplines in biology, from evolutionary genomics genetics to biodiversity conservation. Here, we obtained a highly accurate and nearly complete assembly of the reference genome of the brown alga *Ectocarpus*, a model organism for this key group of eukaryotes. The *Ectocarpus* V5 assembly includes telomeres for most chromosomes and very few gaps and, therefore, provides a new reference genome for the scientific community.

Chromosome folding patterns vary across lineages[56]. For example, in many plant species with relatively large genomes, chromosomes adopt a Rabl configuration during interphase, in which centromere or telomere bundles are associated with opposite faces of the nuclear envelope. For chromosomes with Rabl configuration, their Hi-C maps display a characteristic belt that is perpendicular to the primary diagonal. *Arabidopsis*, in contrast, presents a Rosette configuration[57], where the Hi-C maps feature conspicuous long-range intra-chromosomal contacts due to the formation of megabase-size loops. None of these features were found in the *Ectocarpus* Hi-C map, suggesting that its chromatin adopts a non-Rabl and non-Rosette configuration. The Chromatin arrangement of interphase chromosomes in *Ectocarpus* involved telomeres of all chromosomes and centromeres of all chromosomes clustering together. Therefore, despite different linear genome architectures and centromere sequence compositions, centromere interactions appear to be a pervasive feature in eukaryotes, from plants and animals to brown algae.

TADs, whose boundaries partition the genome into distinct regulatory territories, are a prevalent structural feature of genome packing in animal and plant species, but our observations showed that TADs are not prominent in the *Ectocarpus* genome. Note that *Ectocarpus* has a relatively simple morphology with a reduced number of cell types. The Hi-C maps, thus, are likely to faithfully represent the interphase chromatin structure of male and female *Ectocarpus* rather than an average conformation across multi-cell types as in other more complex organisms. This feature allows us to conclude that *Ectocarpus* has a non-Rabl chromatin conformation and does not exhibit TADs at a local level. The *Arabidopsis* genome is another example in which TADs are absent[26], and this feature is thought to be related to *Arabidopsis* small genome size, high gene density, and short intergenic regions. Given that *Ectocarpus* has similar genomic characteristics, the absence of TADs in *Ectocarpus* supports the hypothesis that TADs may form when the genome size is above a certain threshold[58]. Note that in *Arabidopsis*, despite its genome not having clear TADs, over 1000 TAD-boundary-like and insulator-like sequences were found from Hi-C maps normalized with 2 kb genomic bins[26]. These regions possess similar properties to those of animal TAD borders/insulators, i.e., chromatin contacts crossing insulator-like regions are restricted, and they are enriched for open chromatin. The *Ectocarpus* genome, in contrast, is mainly partitioned in H3K79me2 rich and H3K79me2 poor regions, that largely define A and B compartments, but we did not find any evidence for canonical insulators nor 'TAD-boundery-like' regions. Note that CTCF is absent in the genome of *Ectocarpus*, similar to yeast, *Caenorhabditis elegant,* and plants[59].

The high-resolution, sex-specific Hi-C maps of haploid individuals allowed us to examine the 3D structure of the U and V sex chromosomes in the interphase nucleus of *Ectocarpus* males and females. The U and V chromosomes are largely homomorphic, each containing a small, non-recombining region[9,49] that harbors several dozen genes, including the master male-determining factor *MIN*[60], and a largely heterochromatic landscape[43]. Our *Ectocarpus* V5 yielded gapless SDRs and demonstrated that the U and V SDRs span the centromere. The linkage between the mating type (MT) locus and centromeres is a common feature of haploid MT chromosomes in fungi. For example, in the *Microbotryum* fungi, recombination suppression links the MT-determining loci to centromeres[61,62], and this is thought to help preserve heterozygosity and/or be beneficial under auto-fecundation, as it increases the degree of compatibility between gametes from the same individual. A similar process is unlikely to be operating in *Ectocarpus* because there is no intra-tetrad direct crossing; haploid spores

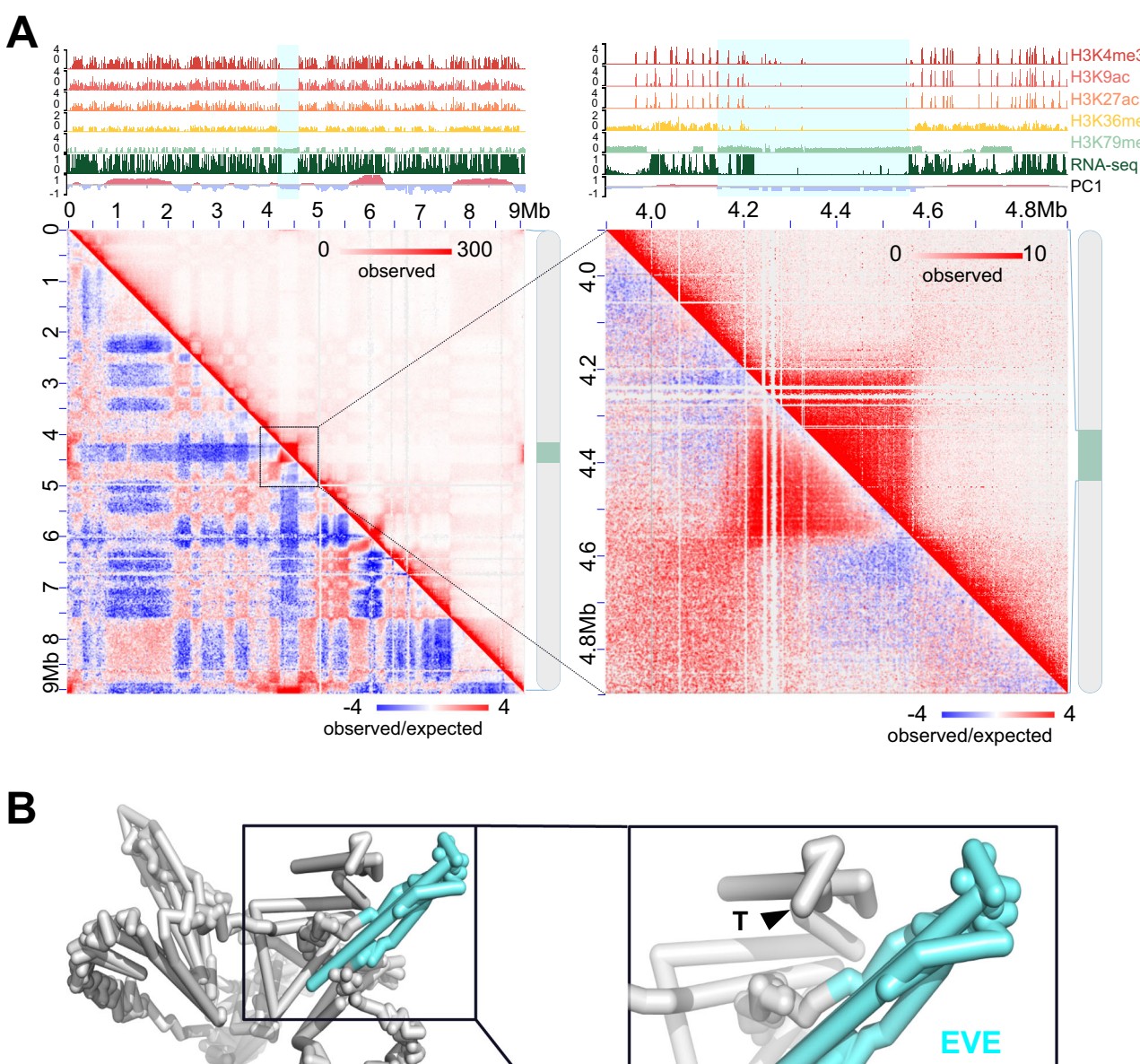

**Fig. 6 | Virus insertion region (EVE) is insulated and shows strong interactions with telomeres. A** Hi-C map of chromosome 6. The zoomed-in region to the right contains the EVE (4.2–4.6 Mb). The tracks above each Hi-C map show A/B compartment annotation (PC1), gene expression (RNA-seq), and histone PTMs ChIP-seq tracks. A blue shade marks the region of the Ec32EVE. **B** Simulated 3D configurations of chromosome 6 at 10 k resolution, generated using 3DMax[105]. The EVE region is colored in aquamarine, and the telomeres are labeled with black triangles. To the right, the Ec32EVE region is zoomed in to highlight the long-range contacts with the telomeres.

disperse after meiosis, develop into male and female gametophytes, and produce gametes at a later stage[63]. Furthermore, only one of the four SDR genes that are contained within the V centromere was sex-linked in the last brown algal common ancestor (Ec-13_001830), while the others became sex-linked later during the expansion of the SDR in the Ectocarpales (Ec-13_001830.1) and in the expansion that is exclusive of *Ectocarpus* (Ec-13_001870.1 and Ec-13_001890.1)[49]. It is, therefore, more conceivable that SDR linkage to the centromere in *Ectocarpus* occurred due to expansion of the non-recombining SDR,

during which the centromere was subsumed in this region likely via a large-scale inversion, as suggested by comparative genomic studies[49]. Nonetheless, we cannot fully exclude a scenario where the ancient SDR genes were initially located in the centromere and were later translocated elsewhere as the SDR expanded.

What is the potential role of the sex chromosome 3D chromatin configuration? Among numerous steps required for gene expression, the spatial organization of the genome is known to modulate DNA accessibility to the transcriptional machinery and to promote contacts

between genes and distant regulatory DNA elements such as enhancers[25]. In the case of *Ectocarpus*, the correct spatial and temporal window of transcriptional activation of genes contained within the SDR is critical to ensure sex determination and differentiation in the brown algal tissues. It is, therefore, likely that the tight transcriptional regulation of the SDR is achieved both by 3D chromatin remodeling in conjunction with histone PTMs and small RNAs[44]. Whilst the 3D chromatin configuration of animal and plant sex chromosomes remains largely elusive, it is well known that chromatin 3D structure is involved in the repression of the silent MT loci in yeast *Saccharomyces cerevisiae* during mating type switching[64,65]. Therefore, it appears that modulation of mating type or sex chromosome architecture may play a significant role in controlling sex-specific features across eukaryotic lineages.

The *Ectocarpus* V5 assembly and high-resolution Hi-C map allowed us to examine centromeric sequences in this organism. We observed 27 unique centromere sequences occurring once per chromosome, a finding that helps to resolve nuclear genome organization and indicates monocentric regional centromeres. The centromeres of *Ectocarpus* may be categorized as transposon-rich and primarily composed of centromere-specific retrotransposons, a relatively common centromeric organization found in species including the amoeba *Dictyostelium discoideum*[66,67] and the oomycete *Phytophthora sojae*[68], where the centromere-specific retrotransposons are associated with the centromere-specific histone H3 variant (cenH3). Although *ECR-1* presumably targets centromeric DNA (and *ECR-2* may have done so in the past), it remains to be determined whether the *ECR* elements constitute the centromere in *Ectocarpus*. The putative centromeres are short relative to the transposon-rich centromeres of many other species[51], and we cannot rule out an association between cenH3 and short AT-rich sequences, as in diatoms[69]. ChIP-sequencing of cenH3 will be required to distinguish between these possibilities.

The fusion of an LTR integrase to a C-terminal chromodomain is most widely known from the evolutionary ancient chromovirus clade of *Metaviridae* LTRs, where the presence of the chromodomain enables recognition of specific histone modifications and targeted insertion at associated genomic sites[70]. In plants, the CRM subclade of chromovirus LTRs contains many centromere-targeting families that accompany satellite arrays and constitute a major component of centromeric DNA[71]. Independent lineages of chromodomain-containing LTRs have been reported in Stramenopiles, including the Chronos *Metaviridae* elements of oomycetes[72] and the *CoDi*-like *Pseudoviridae* (i.e., *Ty1/Copia*) elements of diatoms[73]. Interestingly, *ECR-1* does not appear to be a member of either the chromovirus or Chronos clades and instead is most closely related to chromodomain-containing oomycete LTRs that are yet to be phylogenetically classified (e.g., *Gypsy-20_PR* from *Phytophthora ramorum*). We hypothesize that the chromodomain of *ECR-1* may enable centromere-targeted integration in *Ectocarpus*, either by recognition of cenH3 or other centromere-associated proteins, implying evolutionary convergence with the CRM elements of plants. However, it is unlikely that the targeting mechanism is itself convergent since the centromere-targeting CRM elements feature derived chromodomains that lack the three conserved aromatic amino acids[71], which are present in *ECR-1*.

Viruses that transcribe their DNA within the nucleus have to adapt to the molecular mechanisms that govern transcriptional regulation. The interaction between chromatin and viral-directed modulation of chromatin is a critical component of the viral-host interaction[74]. However, the complexity of the higher-order organization of the host genome and its potential influence in the regulation of gene expression raises questions regarding the spatial arrangement of integrated viral DNA in the host's genome. Phaeoviruses are latent giant double-stranded DNA viruses that insert their genomes into those of their brown algal (Phaeophyceae) hosts[53,55]. Remarkably, although about 50% of individuals in *Ectocarpus* field populations show symptoms of

giant viral infection[75], the *Ectocarpus* strain used in this study has never been observed to produce virus particles, and Ec32EVE genes are transcriptionally silent[31]. Here, we showed that the silencing of Ec32EVE genes correlated with the deposition of large domains of repressive-associated chromatin mark H3K79me2, concomitant with depletion of activation-associated marks. Moreover, the inserted giant viral element was associated with the B compartment, and adopted a highly insulated conformation in the 3D nuclear space, exhibiting strong long-range contacts with the telomeres. It is possible that mechanisms such as phase separation and maybe loop extrusion[76] may underlie the specific 3D configuration of this region. Whilst the detailed mechanisms underlying the relationship between giant virus latency and gene-silencing mechanisms, including the 3D architecture of the chromatin, remain to be determined, our study provides the first description of the 3D configuration of an inserted giant viral element and strong evidence for an interplay between 3D chromatin architecture, H3K79me2 domains, and EVE gene silencing, opening new avenues to gain insights regarding the functional significance of these interactions.

## Methods

### Brown algae culture

Algae were cultured as previously described[77]. Briefly, *Ectocarpus* strains Ec32, Ec25, Ec561, and Ec560 were grown in autoclaved natural seawater (NSW) with PES at 14 °C with the light intensity of 20 μmol photons m$^{-2}$ s$^{-1}$ (12 h light/12 h dark). The medium was changed every week. Before collection, algae were treated with antibiotics: Streptomycin (25 mg/L), Chloramphenicol (5 mg/L), and PenicillinG (100 mg/L) for three days to limit bacterial growth.

### Hi-C

An in situ Hi-C protocol of plants[78] was optimized for brown algae. *Ectocarpus* cultures were collected using a 40 μm filter and fixed in 2% (vol/vol) formaldehyde for 30 min at room temperature, and the cross-linking reaction was quenched with 400 mM glycine. Approximately 50 mg fixed algae suspended in 1 ml nuclei isolation buffer (0.1% triton X-100, 125 mM sorbitol, 20 mM potassium citrate, 30 mM MgCl2, 5 mM EDTA, 5 mM 2-mercaptoethanol, 55 mM HEPES at pH 7.5) with 1X protease-inhibitor in a 2 ml VK05 tube, then homogenized by Precellys Evolution beads homogenizer (Bertin technologies) with the following settings: 7800 rpm, 30 s each time, 20 s pause each grinding cycle, repeat 5 times. Over 1 million nuclei were isolated and digested overnight by Dpn II, DNA ends were labeled with biotin-14-dCTP, then ligated by T4 DNA ligase enzyme. The purified Hi-C DNA was sheared by covaries E220 evolution and libraries were prepared using the NEBNext Ultra II DNA Library Prep Kit (NEB, no. E7645), and the average size of the library was detected by bioanalyzer, the final library was sequenced with 150 bp paired-end reads on an Illumina HiSeq 3000 platform. Two biological replicates were performed for each strain.

### Nanopore sequencing

High molecule weight (HMW) DNA of *Ectocarpus* male (Ec32) and female (Ec25) were isolated using *OmniPrep*™ kit (G-Biosciences) with slight modifications. 500 mg of fresh collected tissue was dried and resuspended in 1 ml lysis buffer, then homogenized using a Precellys mixer. Samples were incubated at 60 °C for 1 h with proteinase K, inversed every 15 min. HMW-gDNA was dried and eluted by 10 mM ph 8.0 Tris-HCl, and incubated at 55 °C for 30 min with 0.5 μL 10 mg/ml RNaseA. The concentration of HMW-gDNA was quantified using an Invitrogen Qubit 4 Fluorometer, and molecule size distributions were estimated using a FEMTO Pulse system (Agilent). The sample was further cleaned and concentrated using AMPure XP SPRI paramagnetic beads (Beckman Colter) at a DNA: bead volume ratio of 1:0.6, followed by two washes using freshly prepared 70% ethanol and resuspension in 10 mM ph 8.0 Tris-HCl. 1 μg HMW-gDNA was used for nanopore library

preparation and sequencing according to the standard protocol of the ONT Ligation Sequencing Kit (Nanopore, https://store.nanoporetech.com/eu/ligation-sequencing-kit110.html). Sequencing was performed on an ONT MinION Mk1B with three R9.4.1 flow cells.

### Re-assembly of genomes assisted by Nanopore and Hi-C

Base-calling was done by ONT Guppy v6.5.7 (--trim_adapters –trim_primers)(Wick et al., 2019). A de novo draft male genome assembly was generated based on Ec32 ONT data by the Canu assembler v2.2(genome Size = 220 m -pacbio-raw)(Koren et al., 2017), with three iterations of error correction by Pilon v1.24[79]. An additional scaffolding step was accomplished by ARCS v1.2.5 ($z = 1500$ $m = 8$-$10000$ $s = 70$ $c = 3$ $l = 3$ $a = 0.3$)[80]. As long read sequencing input, the original ONT read data was extended by the previous assembly[31]; the same strategy was used for the new *Ectocarpus* female draft chromosome 13 with Ec25 ONT reads only mapped to the male chromosome 13 and previous published female SDR scaffold.

The Hi-C raw reads underwent a preprocessing step using Trimmomatic v.0.39 with a default setting to remove the adapters and other Illumina-specific sequences[81]. Subsequently, the clean reads were aligned draft genomes using a 3D de novo assembly (3D-DNA) pipeline, following[82]. The resulting Hi-C contact map, based on the initial chromosomal assembly, was visualized using Juicebox[83]. Juicebox also facilitated the manual adjustment of contig orientations and order along the chromosomes, based on the observed contacts. During this adjustment process, some incorrectly placed sequences were trimmed from the original contigs and reassembled with the appropriate ones. The orientation of the final chromosome name was corrected with the previous reference genome[30]. To refine the assembly, we employed TGS-GapCloser with error correction by racon v1.4.3, along with RFfiller utilizing ONT reads for gap filling[84,85]. Subsequently, an assessment of genome quality was conducted by Benchmarking Universal Single-Copy Orthologs (BUSCO)[34] together with its eukaryote and stramenopiles databases in version odb10.

To be consistent with the V2 genome, we extracted the gapless 1.55 Mb female sex-determining region (SDR) of the female assembly and added it as a separate contig to the male genome (fSDR). This 'reference' assembly is the new *Ectocarpus sp* V5 reference genome.

To identify bacterial contamination in the genome assemblies, the newly assembled scaffolds were analyzed by kraken2 (version 2.1.3)[86], blastn (version 2.13.0, nt database 2022-07-01)[87] and blob tools (version 1.1.1)[88]. Hits identified by all three tools were considered, and corresponding contamination scaffolds were removed. During the contamination analyses, we removed two Hi-C scaffolds corresponding to the bacteria genera *Paraglaciecola* and *Halomonas*.

### Hi-C data analysis

The Hi-C reads were processed using the Juicer pipeline[89], and binning was performed at various sizes, including 2, 5, 10, 20, 50, 100, and 500 kb. The clean Hi-C data was mapped to its corresponding re-assembled reference genome (male or female *Ectocarpus* V5, Supplementary Fig. 1) using Bowtie2[90]. During the alignment, the clean reads were aligned end-to-end, and spanning ligation junctions were trimmed at their 3'-end and realigned to *Ectocarpus* newly assembled genome. The resulting aligned reads from both fragment mates were then paired and stored in a paired-end BAM file. Invalid Hi-C reads, including discarding dangling-end reads, same-fragment reads, self-circled reads, and self-ligation reads, were removed from further analyses.

### Chromosomal contact probability

The reads information processed by the Juicer pipeline in the "merged_nodups.txt" were converted to pairs using the pairix tool[91]. The draft genome was divided into 1000 bp bins, and the contact probability $P(s)$ was calculated and visualized using cooltools[92] following the guidelines provided in the documentation at https://cooltools.readthedocs.io/en/latest/notebooks/contacts_vs_distance.html. In short, $P(s)$ was determined by dividing the number of observed interactions within each bin by the total number of possible pairs.

### A/B compartment identification

The A/B compartment status was determined using Eigenvalues (E1) obtained through eigenvector decomposition of Hi-C contact maps. To calculate the E1 values at a 10 kb resolution, Cooltools software was utilized with the "cooltools eigs-trans" function and GC density file[92]. The resulting E1 values were then loaded into the plaid pattern of Hi-C contact maps. Manual validation based on intra or inter-chromosomal interactions in Hi-C was performed along each chromosome to obtain the final list of "E1" values. Since the direction of eigenvalues is arbitrary, positive values were assigned the label "A", while negative values were assigned the label "B" based on their association with GC or gene density. The compartment border was defined as the edge bin separating the A and B compartments.

### ChIP-seq and RNA-seq

ChIP-seq and RNA-seq data from the male (Ec561) and female (Ec560) strains were obtained from[44]. The datasets include two replicates of H3K4me3, H3K9ac, H3K27ac, H3K36me3, and H3K79me2 samples, as well as two control samples (an input control corresponding to sonicated DNA and anti-histone H3). To process the data, the nf-core ChIP-seq pipeline v2.0.0 was employed[93]. Briefly, the raw data underwent trimming using Trim Galore v0.6.4[94], and the paired-end reads were aligned to the reference genome using BWA v0.7.17[95]. Subsequently, MACS2 with default parameters was used to call broad and narrow peaks[96]. Peaks called with MACS2 for each of the histone marks (normalized by H3) were compared in the A versus B compartment regions to examine the enrichment of each mark per compartment. In short, we used MACS2 to call peaks for each IP, with H3 as the control. The 'fold_enrichment' values from the generated files were used for co-analysis with compartments A/B. The plot values (represented on the $y$-axis) are the log2-transformed 'fold_enrichment' values located in each compartment A or B.

Three replicates of RNA-seq data were trimmed by Trimmomatic v0.39 and mapped on the *Ectocarpus* V5 reference genome (Supplementary Fig. 1) by GSNAP aligner v2021-12-17[81,97], unique mapped read pairs were used to calculate read counts per gene by featureCount v2.0.3, DEseq2 (v1.41.6, Bioconductor) was used for detection differential expression genes with the threshold of adjusted $p$-value = < 0.01 and log2fold change > = 1, TPM (Transcripts Per Million) was used for transcript abundance quantification[43,98,99].

### Centromere characterization

Broad centromeric regions were determined by visually assessing the Hi-C contact maps. To assess the repeat content of these regions, RepeatModeler v2.0.2[100] was run on the male Ec32 V5 genome assembly to generate de novo repeat consensus models, using the flag "-LTRStruct" to perform LTR structural searches. The subsequent repeat library was provided as input to RepeatMasker v4.0.9 (https://www.repeatmasker.org/RepeatMasker/) to identify the genomic coordinates of repeats. Tandem Repeats Finder v4.09.1[101] was run to identify coordinates of satellite and microsatellite DNA using the recommended parameters "2 5 7 80 10 50 2000", enabling satellite DNA with monomers up to 2 kb to be identified. Final tandem repeat coordinates were achieved by combining the simple and low-complexity repeats identified by RepeatMasker with the repeats identified by Tandem Repeats Finder. All other repetitive coordinates identified by RepeatMasker that did not overlap tandem repeats were assumed to be interspersed repeats (i.e., transposable elements).

Putative centromeric repeats were identified by searching for repeat families that were both almost exclusively present in the broad

centromeric regions defined by the contact maps and common to all chromosomes. The two repeat models that met these criteria were then manually curated following Goubert et al.[102]. Retrotransposons related to *ECR-1* were identified by passing the predicted protein to Repbase Censor online tool[103]. Centromeric coordinates were defined as the first to the last copy of *ECR* elements (see Supplementary Data 6). All centromeric analyses were performed on the male Ec32 V5 genome, except for the U chromosome, which was analyzed using the female Ec25 genome.

## Reporting summary

Further information on research design is available in the Nature Portfolio Reporting Summary linked to this article.

## Data availability

The Nanopore and Hi-C data generated in this study have been deposited in the NCBI database under the project number PRJNA1105946. *Ectocarpus* V5 genome and gene annotation, the processed Hi-C, ChIP-seq, and RNA-seq data are available in Edmond of Max Planck Digital Library collection (https://doi.org/10.17617/3. QXUAMN and https://doi.org/10.17617/3.NQDSLW). The RNA-seq and ChIP-seq datasets used in this study were retrieved from the NCBI Gene Expression Omnibus repository: PRJNA1055718.

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

## Acknowledgements

This work was supported by the MPG, the ERC (grant n. 864038), the Moore Foundation (GBMF11489), and the Bettencourt-Schuller Foundation. R.C. is supported by a grant HORIZON-MSCA-2022-PF-01 (Project ID: 101109906). We thank the BMBF-funded de.NBI Cloud within the German Network for Bioinformatics Infrastructure (de.NBI) (031A532B, 031A533A, 031A533B, 031A534A, 031A535A, 031A537A, 031A537B, 031A537C, 031A537D, 031A538A). We thank Remy Luthringer and Andrea Belkacemi for assistance with the algal cultures.

## Author contributions

P.L.: Investigation (lead), Formal analysis (lead), Visualization (lead), Writing – original draft (equal). J.V.: Investigation (equal), Methodology (supporting), Formal analysis (supporting). R.C.: Investigation (equal), Methodology (supporting), Visualization (equal), Formal analysis (equal), Writing – review and editing (supporting). J.B.R. Investigation (equal), Methodology (supporting), Visualization (equal), Formal analysis (equal). E.A. and C.M.: Investigation (supporting). M.B.:Methodology (supporting). F.B.H. and C.L.: Data curation (equal), Visualization (supporting), Formal analysis (equal), supervision (equal), Writing – review and editing (supporting). S.M.C.: Conceptualization (lead), Funding acquisition (lead), Methodology (equal), Project administration (lead), Supervision (lead), Visualization (supporting), Writing – original draft (equal), Writing – review and editing (lead).

## Funding

## Competing interests

The authors declare no competing interests.
