## [Transparent Peer Review file · Nature Communications]

3D chromatin maps of a brown alga reveal U/V sex chromosome spatial organization

Corresponding Author: Dr Susana Coelho

Version 0:

Reviewer comments:

Reviewer #1

(Remarks to the Author)

The study explores the 3D organization of sex chromosomes in the brown alga *Ectocarpus* and its impact on sex determination. Using advanced mapping techniques, the researchers found that *Ectocarpus* chromosomes have a unique non-Rabl conformation with strong telomere and centromere contacts. They lack large interactive domains seen in animals and are regulated by histone modifications. The sex-determining regions on U and V chromosomes are highly insulated and span the centromeres. The study also identifies a unique conformation in a region of chromosome 6 containing an endogenous viral element, raising the question of its functional significance.

The work is of high quality, the findings are original and provide a very valuable resource to the research community. The methodology is sound and the claims are supported. The manuscript is very well written and illustrated.

I have only minor comments

1. L11-12 «.. nuclear three dimensional (3D) folding of chromatin structure is important for gene expression regulation and correct developmental programs..». Although there is a growing number of examples where a causal link between folding and function was indeed established , it still remains to be demonstrated for the majority of cases. In fact, most often, causality was either not addressed, or experiments showed in the contrary that the topology is a consequence, if not a simple correlation, rather than the cause of gene expression. I would thus recommend cautious with this general statement which is unfortunately too often used. Generally, more studies are required to resolve causality between genome topology and function (reviews about genome topology cause or consequence exist).
2. L25-26 "providing insights regarding the functional significance of the chromatin organisation of latent giant dsDNA virus» , do the data really address the functional significance of this specific topological domain created by the viral DNA? Wouldn't it be instead something like "providing insights regarding the impact of latent giant dsDNA virus on the host genome's 3D chromosomal folding"?
3. L38-39 "Moreover, despite the key role that the 3D structure of chromatin plays in gene regulation 17,18" As for point 1, please reformulate (correlation rather than causality), Note that the references 17 and 18 are not appropriate for this statement even if amended, as one is a methodological review and the other address very specific cases in disease. There are a few reviews that would be better suited (eg doi: 10.1016/j.cell.2020.09.014. doi: 10.1016/j.sbi.2023.102723
4. L52 why "plant-type TADs" ? ..lacks typical/canonical TADs?
5. L54 "chromatin loops and compartments are present in *A. thaliana* (e.g. 26)" what is meant with compartments here? A/B compartments? IF yes, a reference for these should be given, for instance the 2 founder papers (one Liu et al, co-authoring this paper and Grob et al.)
6. L60 "sub-nuclear organization of the chromatin structure" not sure to understand what is meant with this. But the paper addresses 3D folding of the genome rather than positioning within a sub-nuclear compartment, thus the term 'sub nuclear' may not be appropriate here?
7. Figure 1 legend: can the authors elaborate on the color legend for contact frequency? Normalized? Z-score? Other metrics? Is it possible to comment on the stripes vs dots on the map? But possibly this figure is not very illustrative at this point of the manuscript since most 3D HiC analyses come in the next section. I suggest to either comment is more or remove it in the first result section.
8. Figure 2: panel A please indicate on the colored legend "relative density of ...?"; panel B: please indicate on the figure which chromosomes were chosen for the illustration
9. L153 "TADs regulate enhancer-promoter contacts and gene expression 37". As for 1, the statement should be moderated

(for instance "in some cases TADs can promote enhancer-promoter contacts important for gene expression")

10. L167-169 "can be identified by applying Principal Components Analysis (PCA) of the correlation heatmap yields the first eigenvector (EV1), (PCA, Lieberman-Aiden et al. 2009)". Please revise the sentence, somehow a subject or verb is missing. The reference is not numbered as the others.

11. Figure 3C: it is really hard to see a difference between the distributions of H3K36 and K79 between the A and B compartments. The mean are the same, the shape of the distributions (violin plots) also. Which Wilcoxon test was used? I am wondering if this test is really appropriate for very large dataset, which inevitably lead to significant differences. As a matter of fact, a large number of loci in the B compartment are enriched in these marks and at a very similar level than loci in the A compartment. Can the authors comment on this and elaborate in the interpretation?

12. L183-184: "and, most conspicuously, H3K36me3, were significantly enriched in the A compartment". Why most conspicuously? And again, related to point 11, given that the B compartment is also very abundant in this mark (same mean, same distribution), the conclusion should be modulated. Perhaps looking at the GO or the K36me3 enriched genes in A and B can provide further material for discussion?

13. Please describe in the text of the result section related to Figure 3, the fold enrichment or fold differences in each mark and gene expression levels between compartments (A vs B)

14. Figure 4 legend : "compartment A/B (PC1)", explain the color code (red=B?, blue=A?)

15. Figure 4 legend : "in the simulated male and female chromosomes", please specify "3D folding simulation" and give a few words about the art of simulation? Model and size of fragments/rods? Reference? – Same for Figure 6 legend

16. L277: is it possible to quantify the "slight enrichment with the H3K79me2 mark"? (fold enrichment and statistical test against control regions)

17. L374-384: the particular folding of the SDR isolating it from neighbouring regions could be discussed in the light of recent concept associating 3D folding and phase separated domains as a joint mean to create specific microenvironment with distinct functionality (eg transcription) ? for instance DOI: 10.1016/j.pbi.2019.03.008 and references therein

18. L388-389 "The centromeres of Ectocarpus may be categorised as transposon-rich and primarily composed of centromere-specific retrotransposons": are there other species with such characteristics? Please comment in the text

19. L420-423- same point as for 17. Link 3D folding and phase-separated microenvironments?

20. Could the authors comment on how frequent the TEL-to-CEN association is found in eukaryotes? And of other giant virus were mapped by HiC in other organisms?

Reviewer #2

(Remarks to the Author)

In their manuscript Liu et al., the sex-specific 3D organization of the model brown alga *Ectocarpus* chromosomes at 2 kb resolution. Authors reported that *Ectocarpus* interphase chromatin exhibits a non-Rabl conformation, with strong contacts among telomeres and among centromeres. Authors proposed that the 3D genome organization of *Ectocarpus* is largely shaped by post-translational modifications of histone proteins that regulate chromatin compaction and mediate transcriptional regulation. Authors also described the spatial conformation and sub-nuclear positioning of the sex determining region (SDR) within the U and V chromosomes and observed that these regions are very insulated and span the centromeres. The results are interesting, but no mechanism is presented in the manuscript to explain the correlation observed. Nothing is shown to prove how important could be the observed 3D genome folding on sex determination or gene expression.

In its present form the paper is an excellent resource but does not provide much conceptual novelty.

Version 1:

Reviewer comments:

Reviewer #1

(Remarks to the Author)

The detailed revisions nicely clarify the points raised during the first evaluation.

I would just like to suggest 3 minor points relative to wording for consideration by the authors (and at the discretion of the editor)

Relative to comment #7 on adding more didactic explanations in the Figure legend 1

Why "diagonal stripes" and not simply stripes? (I see them horizontal and vertical?).

My initial comment was perhaps not so clear, sorry. I meant to elaborate also a bit more in the text, since currently Figure 1B, containing 2 panels with repetitive information with next figures, is commented by a very general sentence "Chromosomal territories were detected in *Ectocarpus*, reflected by strong intra-chromosomal interactions and clear boundaries between chromosomes (see Figure 1B)."

which does not exploit very much these overview HiC maps, hence giving the impression that they are poorly informative. If transferring them to the supplements is not an option then consider explaining the reader a bit more what is the interest of these panels, i.e. the (unique / exciting?) information behind: the *Ectocarpus* genome show similar folding principles as in

other organisms with compartments of genomic regions with high intrachromosomal interactions ("stripes") and loops ("dots") , or more, depending on what the authors find relevant.

Relative to comment #4 on "plant-type TADs".

I understand the motivation of the authors to distinguish plant TADs and animal TADs. But (1) why not plant TADs simply, instead of plant-type TADs (there is no "universal TADs" with a hard-wired checklist of features ?) and (2) I am not sure every reader will be aware of these unique features of plant TADs that the authors refer to in their reply. It may thus be good to drop one or two lines for the reader to explain what distinguish "plant-type TADs" (or, preferably "plant TADs"), from animal TADs.

Relative to comment #17 and 19, on a possible discussion of phase separation behind the SDR structure.

As the authors express the concern to not "overly speculate" about it, they could perhaps replace "It is likely" by "Possibly" ? (and simply saying, "Possibly, mechanisms involving phase separation and loop extrusion may underlie the specific 3D configuration of this region")

REVIEWER COMMENTS

Reviewer #1 (Remarks to the Author):

The study explores the 3D organization of sex chromosomes in the brown alga *Ectocarpus* and its impact on sex determination. Using advanced mapping techniques, the researchers found that *Ectocarpus* chromosomes have a unique non-Rabl conformation with strong telomere and centromere contacts. They lack large interactive domains seen in animals and are regulated by histone modifications. The sex-determining regions on U and V chromosomes are highly insulated and span the centromeres. The study also identifies a unique conformation in a region of chromosome 6 containing an endogenous viral element, raising the question of its functional significance.

The work is of high quality, the findings are original and provide a very valuable resource to the research community. The methodology is sound and the claims are supported. The manuscript is very well written and illustrated.

I have only minor comments

1. L11-12 «.. nuclear three dimensional (3D) folding of chromatin structure is important for gene expression regulation and correct developmental programs.» Although there is a growing number of examples where a causal link between folding and function was indeed established, it still remains to be demonstrated for the majority of cases. In fact, most often, causality was either not addressed, or experiments showed in the contrary that the topology is a consequence, if not a simple correlation, rather than the cause of gene expression. I would thus recommend cautious with this general statement which is unfortunately too often used. Generally, more studies are required to resolve causality between genome topology and function (reviews about genome topology cause or consequence exist).

Re: We thank the reviewer for the overall positive evaluation of our work. We totally agree with the point raised, and we have revised the manuscript accordingly: we have rephrased the sentence (line 11-12) and toned-down descriptions concerning association/causality throughout the manuscript.

2. L25-26 " providing insights regarding the functional significance of the chromatin organisation of latent giant dsDNA virus », do the data really address the functional significance of this specific topological domain created by the viral DNA? Wouldn't it be instead something like "providing insights regarding the impact of latent giant dsDNA virus on the host genome's 3D chromosomal folding"?

Re: We thank the reviewer for the insightful comment; we agree that we should tone down the mention to "functional significance" of this specific topological domain created by viral DNA insertion. We have changed the sentence according to the suggestion (line 23-26).

3. L38-39 "Moreover, despite the key role that the 3D structure of chromatin plays in gene regulation 17,18" As for point 1, please reformulate (correlation rather than causality), Note that the references 17 and 18 are not appropriate for this statement even if amended, as one is a methodological review and the other address very specific cases in disease. There are a few reviews that would be better suited (eg doi: 10.1016/j.cell.2020.09.014. doi: 10.1016/j.sbi.2023.102723

Re: Again, we entirely agree that correlation does not mean causality; we have reformulated the sentence and changed the references as suggested (line 38-41 and line 682-685).

4. L52 why "plant-type TADs" ? ..lacks typical/canonical TADs?

Re: Concerning the use of the term "plant-type TADs" : we wanted to highlight the fact that plant and animals have different types of chromatin structures. In plants, TADs tend to be less distinct and more variable in size compared to the well-defined TADs observed in animals. This difference is often attributed to the unique chromatin organization and regulatory mechanisms in plant cells. Plant-type TADs often exhibit a less pronounced boundary strength and can overlap with each other, unlike the sharp and well-delineated boundaries seen in canonical TADs in animal genomes. By specifying "plant-type TADs," we aimed to acknowledge these unique features and avoid conflating the structures observed in our study with those characterized in animal systems. We would prefer to keep the text as it is, but if the reviewer has a strong feeling against it, we can of course delete this term.

5. L54 "chromatin loops and compartments are present in *A. thaliana* (e.g. 26)" what is meant with

compartments here? A/B compartments? IF yes, a reference for these should be given, for instance the 2 founder papers (one Liu et al, co-authoring this paper and Grob et al.)

Re: By "compartments" we indeed mean A/B compartments. We have revised the manuscript to clarify this point and included the appropriate, suggested references *"However chromatin loops and A/B compartments are present in Arabidopsis (e.g. ^{26,27}) and small structural units within 3D chromatin architecture have been recently described ²⁸."* (line 53-55)

6. L60 "sub-nuclear organization of the chromatin structure" not sure to understand what is meant with this. But the paper addresses 3D folding of the genome rather than positioning within a sub-nuclear compartment, thus the term 'sub nuclear' may not be appropriate here?

Re: We agree with the reviewer's comment and revise it accordingly. We changed the sentence *"this model organism provides the opportunity to investigate the sub-nuclear organization of the chromatin structure of U/V sex chromosomes and compare it to autosomes."* to *"this model organism provides the opportunity to investigate the U/V sex chromosome organization in comparison to autosomes."* (line 59-60).

7. Figure 1 legend: can the authors elaborate on the color legend for contact frequency? Normalized? Z-score? Other metrics? Is it possible to comment on the stripes vs dots on the map? But possibly this figure is not very illustrative at this point of the manuscript since most 3D HiC analyses come in the next section. I suggest to either comment is more or remove it in the first result section.

Re: The color legend represents normalized contact frequency. This normalization process ensures that the color intensity accurately reflects the relative frequency of contacts, making it possible to compare contact frequencies across different regions of the genome.

The diagonal stripes seen in the contact map indicate regions of high contact frequency, corresponding to A/B compartments where regions within the same compartments interact more frequently with each other than with regions in other compartments. The dots scattered across the contact map represent specific loci that have higher contacts than the rest of the genome, often suggesting interactions between telomeres and centromeres or loops between these regions. We have now detailed this in the legend of the figure (line 130-135).

This figure provides an initial overview of the 3D genome organization, although more detailed analyses and interpretations are indeed presented in the subsequent sections of the manuscript. Because the HiC map helped us to assemble the genome (which is the focus of the first section of the results) we thought it would be important to present this figure early in the manuscript. For the sake of the narrative, we would therefore prefer to keep the figure where it is, and as the reviewer suggests, explain better in the legend – we have done so in the revised manuscript

8. Figure 2: panel A please indicate on the colored legend "relative density of ...?"; panel B: please indicate on the figure which chromosomes were chosen for the illustration

Re: We thank the reviewer for pointing this out. We have clarified these points in the legends of the revised version (line 164-169).

Concerning panel B: Instead of showing specific chromosomes, Figure 2B depicts the results of the aggregate chromosome analysis (ACA). In this method, individual chromosomes are linearly transformed to have the same length, and the centromere is placed at the center. The adjusted chromosomes are subsequently used to compute average intra- and interchromosomal contacts. This approach is commonly used to detect chromosome territories in the data (Hoencamp et al., Science 2021). We added an explanation to the legend of the figure (line 168-169).

9. L153 "TADs regulate enhancer-promoter contacts and gene expression ³⁷". As for 1, the statement should be moderated (for instance "in some cases TADs can promote enhancer-promoter contacts important for gene expression")

Re: We agree, we have revised the statement as suggested: *"In some cases, TADs can promote enhancer-promoter contacts important for gene expression³⁸."* (line 159-160)

10. L167-169 "can be identified by applying Principal Components Analysis (PCA) of the correlation

heatmap yields the first eigenvector (EV1), (PCA, Lieberman-Aiden et al. 2009)". Please revise the sentence, somehow a subject or verb is missing. The reference is not numbered as the others.

Re: The sentence has been revised and the reference: "*A/B compartments, which generally correlate to active and repressed chromatin, respectively, can be identified with the first eigenvector (EV1) generated from principal components analysis of the correlation heatmap (PCA)*"³⁵" (line 175-177)

11. Figure 3C: it is really hard to see a difference between the distributions of H3K36 and K79 between the A and B compartments. The mean are the same, the shape of the distributions (violin plots) also. Which Wilcoxon test was used? I am wondering if this test is really appropriate for very large dataset, which inevitably lead to significant differences. As a matter of fact, a large number of loci in the B compartment are enriched in these marks and at a very similar level than loci in the A compartment. Can the authors comment on this and elaborate in the interpretation?

Re: We employed a two-sample Wilcoxon rank sum test to test the log₂ fold change differences between A and B compartments for all histone PTMs marks. We chose to use a Wilcoxon test because the data here is not normally distributed. The reviewer is correct that the differences between the means are subtle but they are nonetheless statistically significant. We have now changed the text to highlight this result (lines 196-205), and we have also added in the figure the exact p-values. We don't have a clear biological explanation for the fact that the enrichment in H3K79me₂ in compartment B is less pronounced than the activation marks in compartment A. Our previous work into chromatin organisation and gene expression in the *Ectocarpus* genome suggests that very little genes are repressed in the strict sense seen in animals and plants, and it appears that the absence of activation marks (and not the presence of "repressive" marks) seems to associate more strongly with changes in gene expression (Gueno et al, NAR, 2023). Note that *Ectocarpus* does not have canonical repressive marks found in most animals and plants (DNA, H3K27 and H3K9 methylation), with only H3K79me₂ behaving the most closely like a repressive mark (Cock et al, Nature 2010; Bourdureau et al, Genome Biol 2022). We feel that a deeper discussion on the nature of repression in these genomes falls outside the scope of the manuscript, since we have already dealt with this in other work, but we are happy to do so if the reviewer thinks this is an important aspect.

12. L183-184: "and, most conspicuously, H3K36me₃, were significantly enriched in the A compartment". Why most conspicuously? And again, related to point 11, given that the B compartment is also very abundant in this mark (same mean, same distribution), the conclusion should be modulated. Perhaps looking at the GO or the K36me₃ enriched genes in A and B can provide further material for discussion?

Re: We agree that the term 'conspicuous' is not well used – we have removed it. We now mention in the text that there is a subtle, albeit significant, difference in enrichment (line 194-197). Moreover, as suggested by the reviewer, we have performed a GO term enrichment analysis on the genes enriched with H3K36me₃. The results are presented in **Reviewer Table 1**. Unfortunately, this analysis did not reveal, in our view, any meaningful material for discussion. We should note that brown algae are distantly related to any other model eukaryotes, so functional analysis of this type remain speculative. Therefore, we don't think this table will add key information to the results section, and we would suggest not to include it, but we are of course happy to do so if the reviewer think this is an important aspect.

Line 192-195: "*such as H3K4me₃, H3K9ac, H3K27ac and H3K36me₃, were significantly enriched in the A compartment (Figure 3C) although the more modestly so for H3K36me₃. Conversely, peaks of H3K79me₂, a histone mark associated with repressed chromatin in Ectocarpus*"^{41,42}, was subtly, but significantly, enriched in the B compartment (Figure 3C)."

13. Please describe in the text of the result section related to Figure 3, the fold enrichment or fold differences in each mark and gene expression levels between compartments (A vs B)

Re: We now mention more clearly that the differences in enrichment of peaks of K36 and K79 in the A compartment genes in relation to B compartment genes are subtle, albeit significant. We also explain how the analysis illustrated in Fig 3C was performed in the legend and methods section. In short, we used MACS2 to call peaks for each IP, with H3 as the control. The 'fold_enrichment' values from the generated files were used for co-analysis with compartment A/B. The plot values (represented on the y-axis) are the log₂-transformed 'fold_enrichment' values located in each compartment A or B. Explanations on how the transcript abundance was calculated are in the methods section (line 631-635).

14. Figure 4 legend : “compartment A/B (PC1)” , explain the color code (red=B?, blue=A?)

Re: The color code in the plot represents the chromosomal compartments identified by Principal Component 1 (PC1): Red indicates positive values corresponding to Compartment A. Blue indicates negative values corresponding to Compartment B. This color scheme highlights the segregation of genomic regions into distinct compartments based on their interaction patterns. These details have been added to the legend of the figure. *“PC1, red indicates positive values corresponding to compartment A, and blue indicates negative values corresponding to compartment B”* (line 247-248).

15. Figure 4 legend : “in the simulated male and female chromosomes” , please specify “3D folding simulation” and give a few words about the art of simulation? Model and size of fragments/rods?

Reference? – Same for Figure 6 legend

Re: We revised the legends for Figures 4 and 6, as requested, incorporating the details about the 3D folding simulation and adding the reference.

“Hi-C map and simulated 3D configurations of sex chromosomes at 10k resolution, employing a maximum likelihood approach, chromosomal structures were constructed from Hi-C data with a default setting of 3D Max⁵⁰.”(line 256-258)

“Simulated 3D configurations of chromosome 6 at 10k resolution, generated using 3DMax⁵⁰” (line 338)

16. L277: is it possible to quantify the “slight enrichment with the H3K79me2 mark”? (fold enrichment and statistical test against control regions)

Re: During the review of our manuscript, we decided to re-evaluate our chromatin analysis of the *Ectocarpus* centromeres. In the previous version, we had normalized our ChIP-seq data with an H3 IP, but on consideration we realised that a total input control is more appropriate given that the centromeres are also composed of centromeric H3 (which would not be recognized by the commercial H3 antibody we used). We have thus re-analysed our ChIP-seq data specifically for this section. Our new analysis now further confirms that the centromeres do indeed have higher levels of H3K79me2 than other chromosomal regions, which can be observed as a positive fold-enrichment signal solely over the centromeric regions. This is not observed for any other histone mark. Importantly, these metaplots (**Fig. 5D**) also include shading that represents the standard error of the averaged ChIP-seq signals, which are very narrow for H3K79me2, confirming the robust and reproducible enrichment across multiple centromeres. For transparency, a heatmap of the chromatin state of each centromere is further shown in **Fig. S10**. We have edited the text in the manuscript accordingly to reflect these new changes.

Furthermore, as requested, we have performed a statistical analysis to compare H3K79me2 levels in centromeres relative to chromosome arms, which has indeed confirmed statistically higher levels in the former (*please see the **boxplot below** showing the enrichment in H3K79me2 mark, p-values represent Wilcoxon tests*). We have chosen not to include this in the manuscript since we have already included a heatmap showing the reproducibility of H3K79me2 levels on each centromere. (but are happy to add the plot in supplemental data if the reviewer thinks this is a important point).

17. L374-384: the particular folding of the SDR isolating it from neighbouring regions could be discussed in the light of recent concept associating 3D folding and phase separated domains as a joint mean to create specific microenvironment with distinct functionality (eg transcription) ? for instance DOI: 10.1016/j.pbi.2019.03.008 and references therein

Re: We agree this is a very interesting point. We would prefer not to overly speculate on the detailed mechanisms underlying the Ectocarpus SDR 3D organization, given this is still early days in our investigations using this emerging model organism. We have added a sentence to suggest these ideas, and specifically for the EVE region which is markedly insulated with a very unique folding- see also our response to comment n.19.

Line 442-450: "It is likely that mechanisms such as phase separation and maybe loop extrusion⁵⁹ underly the specific 3D configuration of this region."

18. L388-389 "The centromeres of Ectocarpus may be categorised as transposon-rich and primarily composed of centromere-specific retrotransposons": are there other species with such characteristics? Please comment in the text

Re: Yes, this is perhaps the most common organization among species that have transposon-rich centromeres, as opposed to short AT-rich regional centromeres or larger satellite-rich centromeres (see cited review by Talbert & Henikoff, 2020). In many species the centromere-specific retrotransposon is directly associated with cenH3. Examples include the amoeba *Dictyostelium discoideum* (centromere-specific DIRS retroelement), the oomycete *Phytophthora sojae* (a *Copia* LTR), and the green alga *Chlamydomonas reinhardtii* (an *L1* LINE, although here the evidence for association between this retrotransposon and cenH3 is currently limited). By some definitions *Drosophila melanogaster* also belongs in this category, since cenH3 is associated with regions featuring a centromere-specific *Jockey* LINE, although the organisation is more complicated due to the presence of large satellite arrays. However, this is not always the case for transposon-rich centromere, e.g. *Neurospora crassa* centromeres are generally enriched in transposons without any obvious centromere-specific component. We thank the reviewer for the opportunity to expand on this point, and have now made reference to the *D. discoideum* and *P. sojae* studies. (line 410-415).

19. L420-423- same point as for 17. Link 3D folding and phase-separated microenvironments?

Re: We have added a sentence to discuss this possibility and included the suggested reference.

Line 435-436: "It is likely that mechanisms such as phase separation and maybe loop extrusion⁵⁹ underly the specific 3D configuration of this region."

20. Could the authors comment on how frequent the TEL-to-CEN association is found in eukaryotes? And of other giant virus were mapped by HiC in other organisms?

Re: The sentence “*Chromatin arrangement of interphase chromosomes in Ectocarpus involved telomeres of all chromosomes and centromeres of all chromosomes clustering together.*” perhaps misled the reviewer – we do not report TEL-to-CEN contacts, but all telomeres cluster together (TEL-to-TEL) and all centromeres cluster together (CEN-to-CEN) (not telomeres + centromeres all together). Concerning the giant virus results, as far as we know this is the first description of a giant virus being mapped by Hi-C. We think this is actually an important point that we have not perhaps emphasized sufficiently - and we now have added a sentence to highlight this novelty.

Line 445: “*our study provides the first description of the 3D configuration of an inserted giant viral element and strong evidence for an interplay between 3D chromatin architecture, H3K79me2 domains and EVE gene silencing, opening new avenues to gain insights regarding the functional significance of these interactions.*”

Reviewer #2 (Remarks to the Author):

In their manuscript Liu et al., the sex-specific 3D organization of the model brown alga *Ectocarpus* chromosomes at 2 kb resolution. Authors reported that *Ectocarpus* interphase chromatin exhibits a non-Rabl conformation, with strong contacts among telomeres and among centromeres. Authors proposed that the 3D genome organization of *Ectocarpus* is largely shaped by post-translational modifications of histone proteins that regulate chromatin compaction and mediate transcriptional regulation. Authors also described the spatial conformation and sub-nuclear positioning of the sex determining region (SDR) within the U and V chromosomes and observed that these regions are very insulated and span the centromeres. The results are interesting, but no mechanism is presented in the manuscript to explain the correlation observed. Nothing is shown to prove how important could be the observed 3D genome folding on sex determination or gene expression.

In its present form the paper is an excellent resource but does not provide much conceptual novelty.

Re: We are pleased that Reviewer 2 find our results interesting and agrees that our work is an excellent resource for the community. While the nature of our work here is largely descriptive, as are most studies that look into 3D genome folding, we would like to reiterate and emphasise that this nonetheless delivers new conceptual insights on several levels.

Firstly, this is the first report of 3D genome organization in brown algae, a lineage with independent evolutionary origins from both plants and animals. Our work is thus of important value to the biological community, not least as it helps unify our understanding of how chromatin is organized in distinct multicellular lineages of eukaryotes. Following on from this, we have previously reported how the chromatin landscape in brown algae completely differs with all other lineages given the absence of DNA methylation, H3K9 methylation as well as PRC2-mediated H3K27me3. It is thus informative to understand how the lack of these ubiquitous epigenetic modifications can affect chromatin organisation in a complex eukaryotic organism.

Secondly, our work is the first to show how the insertion of a giant DNA virus can affect chromatin topology in eukaryotes, which is timely since it is becoming increasingly evident that giant DNA virus insertions are more common in eukaryotic genomes than previously thought. By examining both chromatin modifications and 3D genomic architecture, our data has thus broad implications on host chromosomal folding and functional outcomes that underscore the complexity of host-virus interactions. Our Hi-C data reveals that the viral DNA insertion creates a distinct topological domain within the host genome. This new domain appears to be insulated from neighboring regions, suggesting that the viral insertion affects the overall 3D chromosomal architecture. Our histone PTM and RNA-seq profiling clearly show that the viral DNA inserted on chromosome 6 is highly silenced, and is enriched with the histone mark H3K79me2, which appears to function as a repressive chromatin mark in *Ectocarpus*. Interestingly, we show that the giant DNA viral insertion forms a distinct topological domain that may have several functional implications. One the one hand, its insertion appears to have helped establish a repressive chromatin environment that likely not only affects the viral genes but also nearby host genes, thereby altering the host's gene expression profile. The

integration of viral DNA and the subsequent impact on chromatin topology is likely to also impact genomic stability, potentially affecting DNA repair mechanisms and susceptibility to genomic rearrangements. Note that the points raised above are discussed in the manuscript but we have been very careful not to be over-speculative. We are happy to reinforce these ideas in the discussion section if the editor and reviewer think these are important points to emphasize further.

Thirdly, we of course also address the 3D conformation of UV sex chromosomes, which to our knowledge is the first description of how sex chromosomes of any type are organised in a complex eukaryotic organism.

REVIEWERS' COMMENTS

Reviewer #1 (Remarks to the Author):

The detailed revisions nicely clarify the points raised during the first evaluation.

I would just like to suggest 3 minor points relative to wording for consideration by the authors (and at the discretion of the editor)

Relative to comment #7 on adding more didactic explanations in the Figure legend 1

Why "diagonal stripes" and not simply stripes? (I see them horizontal and vertical?).

RE: We thank the reviewer for the comment; we have replaced "diagonal stripes" by "stripes" according to the suggestion.

My initial comment was perhaps not so clear, sorry. I meant to elaborate also a bit more in the text, since currently Figure 1B, containing 2 panels with repetitive information with next figures, is commented by a very general sentence "Chromosomal territories were detected in *Ectocarpus*, reflected by strong intra-chromosomal interactions and clear boundaries between chromosomes (see Figure 1B)." which does not exploit very much these overview Hi-C maps, hence giving the impression that they are poorly informative. If transferring them to the supplements is not an option then consider explaining the reader a bit more what is the interest of these panels, i.e. the (unique / exciting?) information behind: the *Ectocarpus* genome show similar folding principles as in other organisms with compartments of genomic regions with high intrachromosomal interactions ("stripes") and loops ("dots") , or more, depending on what the authors find relevant.

RE: We thank the reviewer for the insightful comment, and we agree that a more detailed explanation of Figure 1B is useful to highlight the significance of the Hi-C maps for male and female *Ectocarpus*. We have added the following text:

"The global Hi-C maps of male and female Ectocarpus show no noticeable differences among autosomes, suggesting that the overall chromosomal territory organization is highly similar between the sexes. Additionally, both the male and female sex chromosomes do not display any distinct intra- or inter-chromosomal contact patterns that differentiate them from autosomes. Therefore, the Ectocarpus genome folding on a chromosomal level appears to be consistent across both sexes and all chromosomes."

Relative to comment #4 on "plant-type TADs".

I understand the motivation of the authors to distinguish plant TADs and animal TADs. But (1) why not plant TADs simply, instead of plant-type TADs (there is no "universal TADs" with a hard-wired checklist of features ?) and (2) I am not sure every reader will be aware of these unique features of plant TADs that the authors refer to in their reply. It may thus be good to drop one or two lines for the reader to explain what distinguish "plant-type TADs" (or, preferably "plant TADs"), from animal TADs.

RE: We thank the reviewer for this helpful comment and agree that clarification is needed. We propose to address both points as follows:

1. We modified the words from "plant-type TADs" to "plant TADs" as suggested.

2. To provide clarity for readers unfamiliar with the distinctions between plant and animal TADs, we added a brief explanation in the manuscript:

“In contrast to animal TADs that have sharp and well-delineated boundaries on Hi-C maps, plant TADs exhibit a less pronounced boundaries due to weaker chromatin insulation.”

Relative to comment #17 and 19, on a possible discussion of phase separation behind the SDR structure.

As the authors express the concern to not "overly speculate" about it, they could perhaps replace "It is likely" by "Possibly" ?

(and simply saying, "Possibly, mechanisms involving phase separation and loop extrusion may underlie the specific 3D configuration of this region")

RE: We agree with the reviewer's comment and revise it according to the suggestion.